# Uncovering the dynamics and consequences of RNA isoform changes during neuronal differentiation

Jelena Ulicevic[1,2,8], Zhihao Shao[1,3,8], Olga Jasnovidova[1,8], Annkatrin Bressin [1], Martyna Gajos[1,3], Alex HM Ng[4,5], Siddharth Annaldasula[1], David Meierhofer[6], George M Church[4,5], Volker Busskamp[7] & Andreas Mayer [1]✉

## Abstract

Static gene expression programs have been extensively characterized in stem cells and mature human cells. However, the dynamics of RNA isoform changes upon cell-state-transitions during cell differentiation, the determinants and functional consequences have largely remained unclear. Here, we established an improved model for human neurogenesis in vitro that is amenable for systems-wide analyses of gene expression. Our multi-omics analysis reveals that the pronounced alterations in cell morphology correlate strongly with widespread changes in RNA isoform expression. Our approach identifies thousands of new RNA isoforms that are expressed at distinct differentiation stages. RNA isoforms mainly arise from exon skipping and the alternative usage of transcription start and polyadenylation sites during human neurogenesis. The transcript isoform changes can remodel the identity and functions of protein isoforms. Finally, our study identifies a set of RNA binding proteins as a potential determinant of differentiation stage-specific global isoform changes. This work supports the view of regulated isoform changes that underlie state-transitions during neurogenesis.

**Keywords** Cell Differentiation; Gene Expression; Multi-omics; Nanopore Sequencing; RNA Isoforms
**Subject Categories** Chromatin, Transcription & Genomics; Neuroscience; RNA Biology

## Introduction

During differentiation, cells start in a pluripotent state and differentiate into different mature cell types. Pluripotent stem cells and fully differentiated cells have been extensively studied representing the start and end points of the differentiation process, respectively (Abascal et al, 2020; Mallon et al, 2013). The dynamic cell-state-changes that link the pluripotent with the fully differentiated mature state are less well characterized. A major improvement in the understanding of dynamic cell-state-transitions was enabled by the availability of in vitro differentiation models that can recapitulate state-changes in vivo (Mertens et al, 2016; Liu et al, 2020). Mammalian cell differentiation systems are now available to generate many cell types including neurons (Ng et al, 2021; Flitsch et al, 2020; Pawlowski et al, 2017). The vast majority of neuronal differentiation models are of murine origin. Well-defined human neuronal cell differentiation systems are still lacking.

More recently, induced expression of cell fate-determining transcription factors in induced pluripotent stem cells (iPSCs) has been used to initiate rapid and efficient neural differentiation (Boyer et al, 2012; Matsushita et al, 2017; Busskamp et al, 2014; Tsunemoto et al, 2018; Ng et al, 2021; Zhang et al, 2013; Pawlowski et al, 2017). The main advantages of these model systems are that (i) neurogenesis is achieved under defined growth conditions (Matsushita et al, 2017; Ng et al, 2021), (ii) the cells pass through neural progenitor-like stages (Busskamp et al, 2014; Ng et al, 2021; Kutsche et al, 2018), and (iii) they are amenable to 'omics' approaches that reveal systems-wide gene expression changes during neurogenesis. Most previous studies analyzing the dynamics of gene expression regulation during differentiation have mainly focused on transcription factor (TF) binding, transcriptional and epigenetic changes (Tsankov et al, 2015; Dixon et al, 2015; Arner et al, 2015; Farlik et al, 2016; Krendl et al, 2017; Velasco et al, 2017; Appel et al, 2021). The dynamic alterations in RNA processing during human neurogenesis, such as alternative splicing, have remained unclear.

Alternative splicing of pre-mRNA has emerged as an important player in cell differentiation and neurogenesis (Fiszbein and Kornblihtt, 2017; Furlanis and Scheiffele, 2018; Raj and Blencowe, 2015). Alternative RNA splicing generates many transcript isoforms from a single gene in human cells, greatly expanding the coding potential of the human genome (Marasco and Kornblihtt, 2023; Djebali et al, 2012). Transcripts from 95% of multi-exon genes undergo alternative splicing (Pan et al, 2008; Wang et al, 2008). There are several modes of alternative splicing. A common

[1]Otto-Warburg-Laboratory, Max Planck Institute for Molecular Genetics, Berlin, Germany. [2]Department of Biology, Chemistry and Pharmacy, Freie Universität Berlin, Berlin, Germany. [3]Department of Mathematics and Computer Science, Freie Universität Berlin, Berlin, Germany. [4]Department of Genetics, Blavatnik Institute, Harvard Medical School, Boston, USA. [5]Wyss Institute for Biologically Inspired Engineering at Harvard University, Boston, USA. [6]Mass Spectrometry Facility, Max Planck Institute for Molecular Genetics, Berlin, Germany. [7]Department of Ophthalmology, University Hospital Bonn, Medical Faculty, Bonn, Germany. [8]These authors contributed equally: Jelena Ulicevic, Zhihao Shao, Olga Jasnovidova. ✉E-mail: mayer@molgen.mpg.de

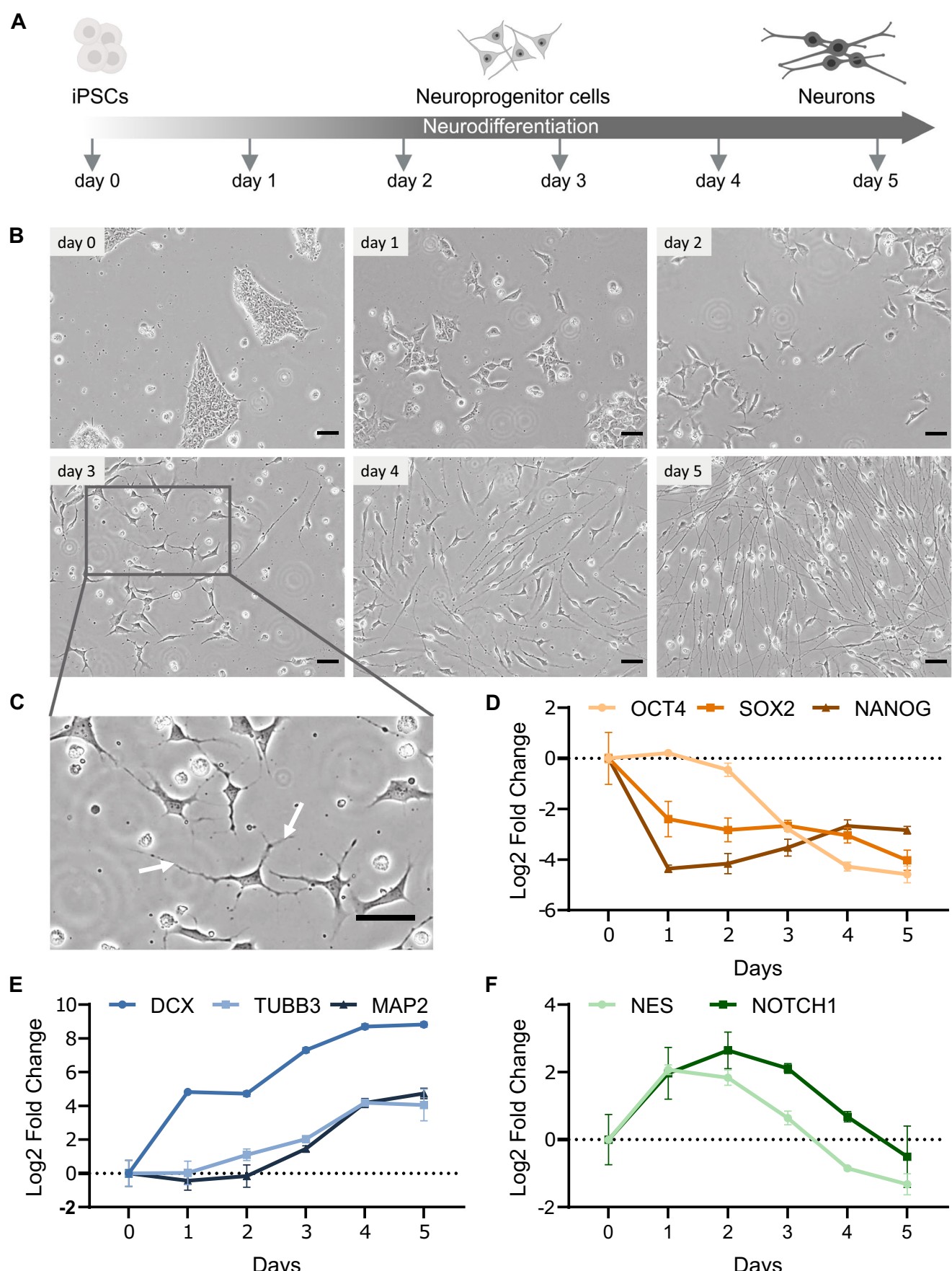

◀ **Figure 1.** **Neuronal differentiation upon induced expression of Neurogenin-3.**

(**A**) Scheme of neuronal differentiation from induced human pluripotent stem cells (iPSCs) upon doxycycline-induced NGN3 expression. (**B**) Phase contrast microscopy images showing cell morphological changes during the differentiation time course. Scale bars indicate 50 μm. (**C**) Zoomed-in view of microscopy image obtained after 3 days of NGN3 induction. White arrows point to selected emerging neurites. Scale bars indicate 50 μm. (**D–F**) Expression of pluripotency markers (**D**), neuronal markers (**E**) and neural progenitor markers (**F**) during neuronal differentiation as measured by RT-qPCR ($n = 3$). Data information: In (**D–F**), data is presented as mean ± SD. Horizontal bars represent the mean of the log2 fold change of the normalized expression compared to day 0 of three biological replicate measurements. Error bars represent the corresponding standard deviation (SD) of the log2 transformed fold change. Source data are available online for this figure.

mode in mammalian cells is exon skipping, in which an exon of interest is included or excluded from the mature RNA. Alternative splicing is regulated by short *cis*-motifs known as splicing enhancers and silencers that are bound by RNA binding proteins (RBPs) (Raj and Blencowe, 2015). Most of the *cis*-regulatory motifs involved in alternative splicing are located within ~300 nucleotides of splice sites (Raj and Blencowe, 2015). The combinatorial control by *cis*-motifs and splicing regulatory RBPs has been proposed to represent a 'splicing code' (Wang and Burge, 2008). It is debated what fraction of splice-isoforms have function and are converted into protein-isoforms (Tress et al, 2017; Blencowe, 2017), but there is clear evidence that alternative splicing can diversify the proteome and affect RNA stability (Weatheritt et al, 2016; Kim et al, 2014; Floor and Doudna, 2016; Maier et al, 2020; Pelechano et al, 2013).

Alternative splicing is particularly prevalent in the mammalian nervous system, and alternative splicing patterns change during neuronal differentiation (Vuong et al, 2016; Raj and Blencowe, 2015). Mutations in neural RBPs involved in alternative splicing control and alterations in alternative splicing patterns have been associated with neurodevelopmental disorders (Lenzken et al, 2014; Licatalosi and Darnell, 2006).

In addition to alternative splicing, there are other mechanisms that can contribute to transcriptome diversification (de Klerk and 't Hoen, 2015), including the use of alternative transcription start sites (TSSs) (Djebali et al, 2012; Sandelin et al, 2007; Juven-Gershon et al, 2008) and alternative cleavage and polyadenylation (Mitschka and Mayr, 2022; Pereira-Castro and Moreira, 2021). The determinants, dynamics and functional consequences of isoform changes during human neuronal differentiation are not well characterized. Furthermore, the relative contribution of the different mechanisms to the complexity and dynamics of the transcriptome during neurogenesis is not clear.

RNA sequencing (RNA-seq) has been the method of choice for analyzing RNA isoforms in mature and differentiating cells (Stark et al, 2019). More recently, long-read RNA-seq technologies, such as those from Oxford Nanopore Technology (ONT) and Pacific Biosciences (PacBio), have become available that allow end-to-end sequencing of full-length RNAs/cDNAs (Sharon et al, 2013; Bolisetty et al, 2015). Long-read RNA-seq technologies are now widely used and have emerged as the new method-of-choice for transcriptome-wide RNA isoform analysis (Hardwick et al, 2019). However, the relationship between short- and long-read RNA-seq in detecting RNA isoform changes during dynamic cell-state-transitions as they occur during cell differentiation has remained puzzling.

In this systems-wide study, we used a multi-omics approach to investigate the dynamics, functional consequences, and determinants of transcript isoform changes during human neuronal differentiation. By establishing and using an improved human

neurogenesis model, this combined experimental and computational approach reveals that the massive shifts in cell morphology during neurogenesis strongly correlate with dynamic RNA isoform changes. We captured widespread transcript isoform changes at all stages of neurogenesis that dynamically affect the proteome. We also identified thousands of novel RNA isoforms, many of which are expressed in a differentiation stage-specific manner. Finally, this study uncovered a set of RBPs that may drive these widespread isoform changes.

## Results

### An efficient in vitro human neuronal cell differentiation model

To study the fundamental principles that underlie the dynamic gene expression changes during human neuronal cell differentiation, we generated an improved human neuronal differentiation system.

This differentiation model relies on the induced expression of the Neurogenin transcription factor Neurogenin-3 (NGN3) in human induced pluripotent stem cells (iPSCs) (Fig. 1A; Methods). Upon induction of NGN3 expression, cells differentiate into neuron-like cells of mostly bipolar morphology within 5 days. Thereby, cells undergo massive cell morphology changes throughout the differentiation time course (Fig. 1B). A major change that becomes visible on day 3 is the outgrowth of neurites from both sides of the cell body accompanied by a decrease in the size of the cell body (Fig. 1B,C). Cells exit pluripotency between day 0 and 1 as indicated by the decrease in expression of the pluripotency markers, especially of *NANOG* and *SOX2* (Fig. 1D). Neuronal fate is acquired between day 2 and day 5 as shown by an increased expression of the neuronal marker genes, particularly of *DCX* (Fig. 1E). Thereby, cells transition through a progenitor-like phase at around day 2 where peak expression of neural progenitor markers *NES* and *NOTCH1* were detected (Fig. 1F).

We generated a monoclonal NGN3 iPSC line, called NGN3m, and optimized induction so that cells synchronously and efficiently differentiate into neuron-like cells. This is illustrated by the similar morphology of cells observed within different time points during the differentiation course (Fig. 1A,B) and through the uniform change in the expression of marker genes (Fig. 1D–F). The induced NGN3m cells differentiate reproducibly and efficiently in standard stem cell media as indicated by a high yield of NGN3m neurons. >90% of the initially plated iPSCs differentiate into neuron-like cells of nearly identical bipolar morphology (Fig. 1B).

Taken together, we have established a human neuronal differentiation model in which pluripotent stem cells differentiate

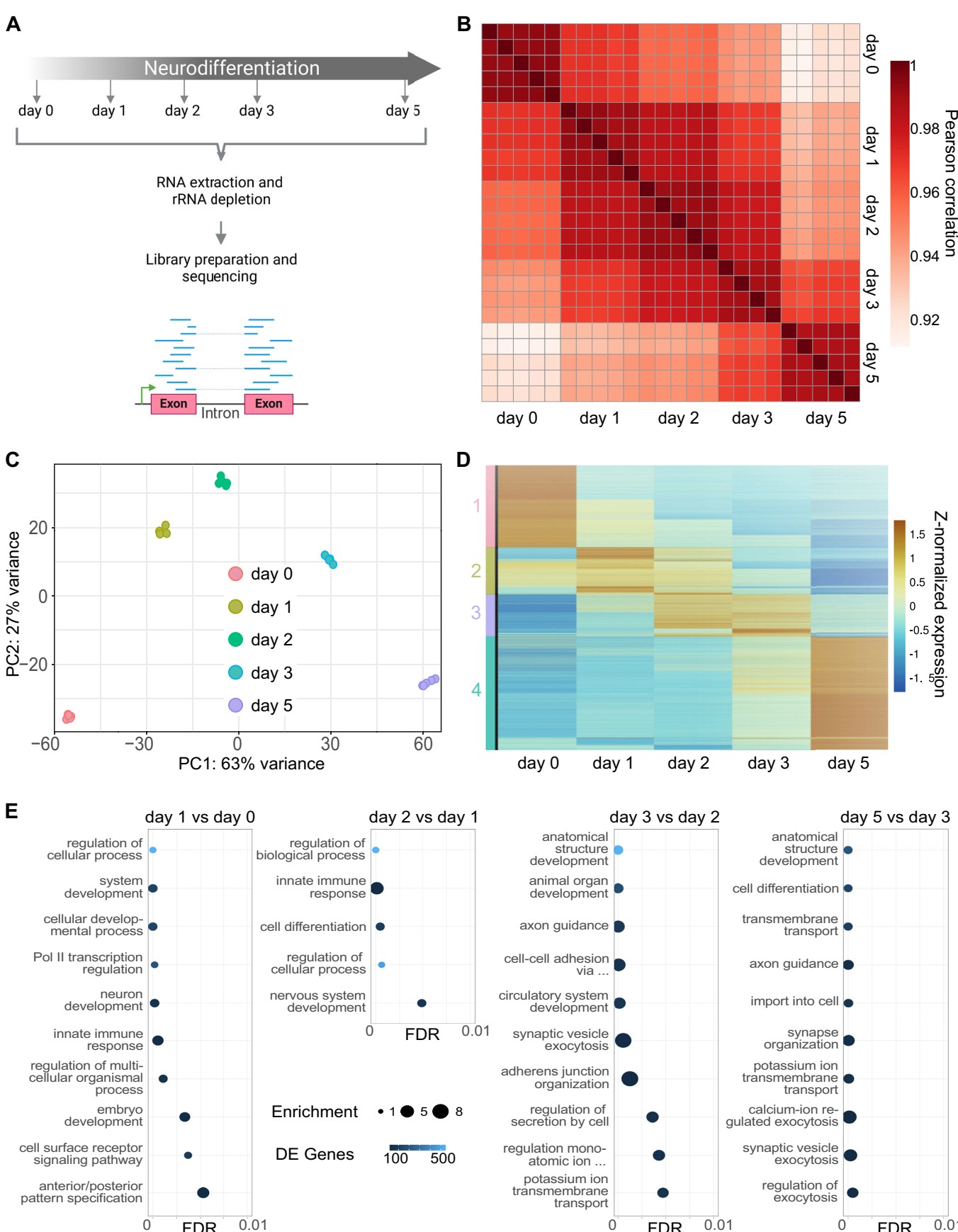

 **Figure 2. Dynamic changes in transcript abundance during human neurogenesis.**

(A) Scheme of the RNA-seq time-course experiment. (B) Pearson's correlation coefficient of RNA-seq data (gene raw counts) as calculated by RSEM for the different differentiation time points. (C) PCA plot of the top 500 most variable expressed genes throughout the differentiation course as measured by RNA-seq. (D) Z-normalized heatmap of all differentially expressed genes ($n = 19,904$) during the differentiation course, including k-mean clustering of genes. The obtained four clusters are color-coded. (E) Top ten significant gene ontology (GO)-slim terms (level 0) from upregulated differentially expressed genes during the differentiation course. A full list of obtained GO-terms is provided in Appendix Fig. S1.

into neuron-like cells through a progenitor stage in a synchronous manner, well suited for the analysis of dynamic state-transitions during neurogenesis.

## Phases of main transcript level changes correlate with shifts in cell morphology

To gain insights into the mechanisms that underlie the strong cell morphology changes during human neurogenesis, we first quantified transcriptome changes at distinct time points during differentiation using RNA-seq (Fig. 2A). The biological replicate measurements strongly correlated (Fig. 2B). The correlation analysis revealed the strongest differences in the transcriptomes of consecutive days between day 0 and day 1, and between day 3 and day 5 of neurogenesis (Fig. 2B) indicating that most expression changes occur during these two developmental phases. The principal component analysis (PCA) of RNA-seq measurements confirmed this finding (Fig. 2C). Accordingly, the first principal component (PC1) that explains 63% of the variation in transcript levels reflects the differentiation time (Fig. 2C). The endpoints of the neuronal differentiation process (day 0 and day 5) were most separated indicating that the corresponding expression programs were most distinct (Fig. 2B,C).

Of note, the strongest transcriptome changes were detected when major morphological shifts occurred namely between day 0 and day 1, during exit of pluripotency at which cells lost stem cell morphology (Fig. 1B). Moreover, stark morphology changes were visible between day 3 and day 5 when cells acquired the neuron shape (Fig. 1B). Analysis of cell cycle marker gene expression provided evidence that cells enter the post-mitotic stage between day 2 and day 3 (Fig. EV1A) of neurogenesis which was consistent with microscopic observations. Expression of genes with a positive role (cluster 1) in cell cycle control declined whereas genes with mainly a negative function (cluster 2) were upregulated between day 2 and day 3 (Fig. EV1A).

To reveal gene sets for which the expression changed similarly between the different developmental time points, we performed a cluster analysis. This analysis uncovered four main groups of genes corresponding to clusters 1, 2, 3, and 4 (Fig. 2D). Genes of cluster 1 or 4 were mainly expressed on day 0 or 5 of differentiation, respectively (Fig. 2D). Genes of cluster 2 were mainly expressed on day 1 and genes of cluster 3 showed peak expression on day 2 and day 3 of the differentiation course (Fig. 2D). A Gene Ontology (GO) enrichment analysis for genes that are differentially expressed between subsequent time points of differentiation revealed an enrichment of terms related to development, such as "system development", "cellular developmental process" and "neuron development", for early time points. For the later time points, we observed an enrichment of terms for neurons and more mature

cell-states, such as "axon guidance", "synapse organization", and "synaptic vesicle exocytosis" (Fig. 2E; Appendix Fig. S1).

We also observed dynamic changes in the expression of transcription factors (TFs; Fig. EV1B). Notably, the set of TFs that showed peak expression levels in the pluripotent state (day 0) and the differentiated state (day 5) was most distinct (Fig. EV1B) indicating changes in regulatory cascades during human neuronal differentiation.

Analysis of neuronal marker gene expression revealed an upregulation of markers for immature, mature and GABAergic neurons during NGN3m differentiation (Fig. EV1C). Finally, we compared the expression profile of the obtained bipolar neuron-like cells with the corresponding profiles available in the Developing Human Brain Atlas (Miller et al, 2014). The highest similarity was detected for neurons of different brain regions of the human embryo, including the cerebral cortex, 9 to 21 weeks post-conception (Fig. EV1D). High similarity in the expression profile was also detected for neurons of the cerebral cortex of more mature brains, such as of individuals between 11 and 36 years of age (Fig. EV1D).

Taken together, these data suggest that most profound changes in transcript levels occur at the early and late stages of the neuronal differentiation course, coinciding with major cell morphology shifts.

## Complex RNA isoform changes often underlie dynamic shifts in total transcript levels

We next examined if the dynamic changes in the total abundance of transcripts during neuronal differentiation were due to alterations in the expression of corresponding transcript isoforms.

We first inferred transcript isoforms from the RNA-seq time-course data (Methods). This analysis revealed 69,344 transcript isoform changes between two or more time points during neurogenesis. Of all expressed genes ($n = 14,655$) with at least two annotated transcript isoforms, 9291 (63%) had at least one isoform change during NGN3m differentiation (Fig. 3A). The majority of these genes showed transcript isoform changes irrespective of whether their overall expression significantly differed or remained similar during the differentiation course (Fig. 3A) indicating that shifts in RNA isoform expression often underlie total gene expression levels. A clear change in the relative abundance of the expression of transcript isoforms could be observed for *DLL3* (Fig. EV2A) which encodes for a Notch ligand and has been implicated in neurogenesis (Lendahl, 1998). In some cases, a switch in the expression of isoforms occurred, where the level of one isoform increases and exceeds the abundance of the other main isoform during the differentiation course. Isoform switching could be detected for the two transcript forms of *PFN2*

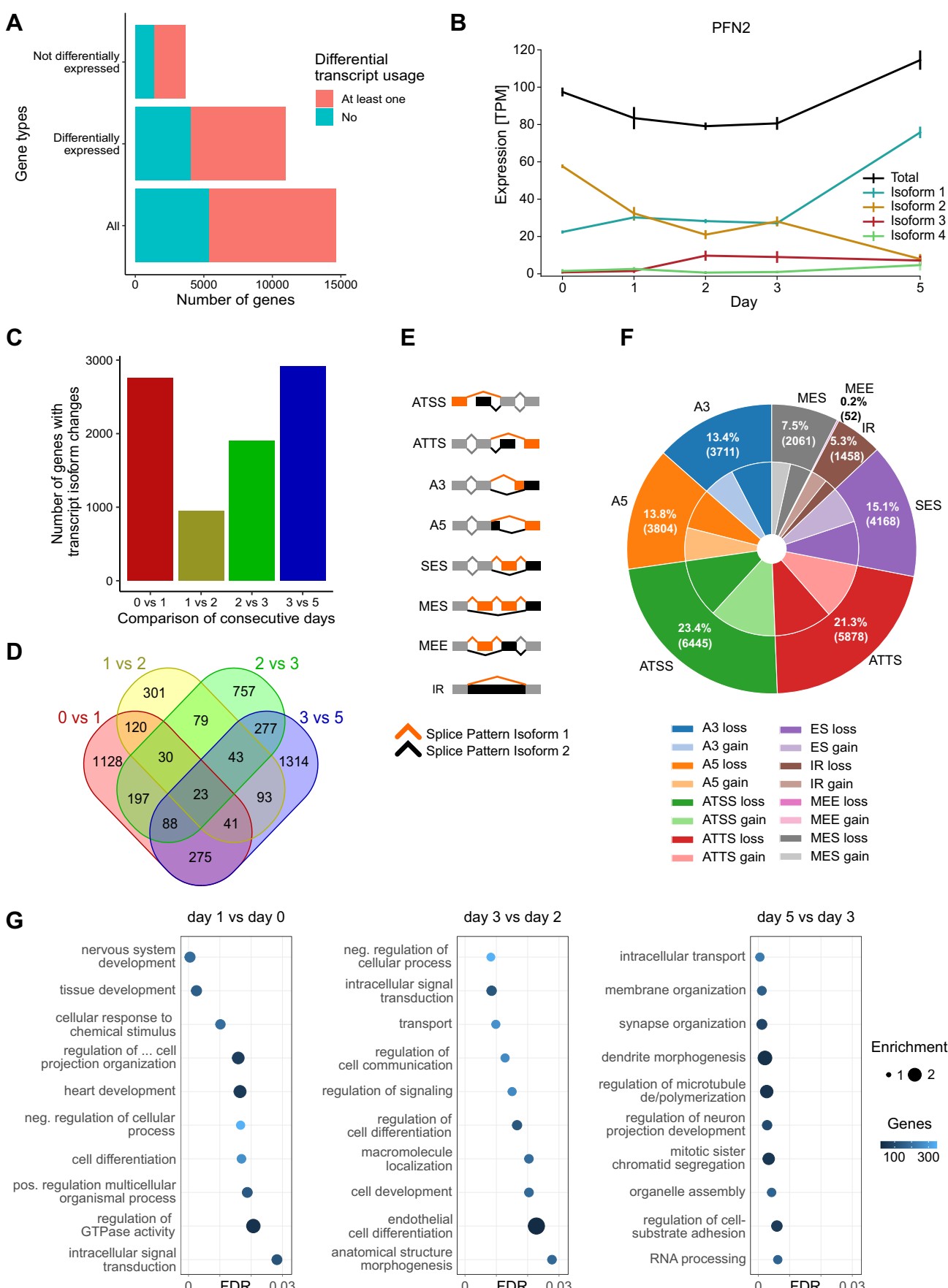

**Figure 3.   Dynamic transcript isoform changes during human neurogenesis.**

(A) Bar plot showing the fraction of genes with at least one RNA isoform change during NGN3m differentiation for different gene types. (B) Quantification of total gene expression of *PFN2* and relative abundance of four transcript isoforms inferred from RNA-seq time-course data. Expression levels were normalized as transcript per million (TPM). (C) Bar plot illustrating the number of genes with changes in isoform expression when comparing consecutive days. (D) Venn diagram depicting the overlap of genes with isoform expression changes for the comparisons between consecutive days. (E) Schematic view of eight main types of alternative transcription and splicing. ATSS: Alternative Transcription Start Site; ATTS: Alternative Transcription Termination Site which in this case means alternative use of polyadenylation sites; A3: Alternative 3′ Splice Site; A5: Alternative 5′ Splice Site; SES: Single Exon Skipping; MES: Multiple Exon Skipping; MEE: Mutually Exclusive Exons; and IR: Intron Retention. (F) Fractions of alternative transcription and splicing mechanisms of differentially expressed isoforms across all consecutive day-to-day comparisons. 'Gain' refers to isoform changes where upregulated RNA isoforms originated from the respective splicing mechanism, while 'loss' denotes cases where upregulated isoforms were no longer associated with the corresponding splicing mechanism (pie chart, center). (G) Top ten significant GO terms (level 0) from genes with differential isoform expression patterns between indicated days of the differentiation course.

that were expressed during human neurogenesis (Fig. 3B). *PFN2* encodes for an actin-binding protein that regulates cytoskeletal dynamics during neuronal differentiation (Birbach, 2008). The shifts in individual isoform levels often sum up to the observed dynamic change in the total expression level of the corresponding gene (Figs. 3B and EV2A).

We detected most isoform changes within the first 24 h and towards the end of the differentiation time course (Fig. 3C). There was only a modest overlap of isoform changes between the individual time points. The vast majority of isoform changes were specific to the respective stage of differentiation (Fig. 3D).

The majority (55%) of isoform changes arose from alternative splicing including exon skipping and intron retention (Fig. 3E,F). A substantial fraction (45%) originated from the usage of alternative transcription start and polyadenylation (pA) sites (Fig. 3E,F). A GO term analysis for genes encoding transcripts that showed isoform changes during differentiation revealed enrichment of terms related to neuronal cell development, differentiation, cell communication, RNA processing and processes that underlie cell morphology changes including microtubule regulation (Figs. 3G and EV2B).

Together, this data indicates widespread and complex transcript isoform changes during human neuronal differentiation that often underlie total shifts in transcript levels.

## Nanopore sequencing directly unveils transcript isoforms during neurogenesis

The time-course RNA-seq experiments indicated the prevalence of isoform changes during neurogenesis (Fig. 3A–F). However, this data has intrinsic limitations from short-read sequencing due to the short read-length (100 nt). Changes in the expressed isoforms cannot directly be identified because reads are too short to span consecutive exon-exon junctions. The quantification of expressed isoforms is merely estimated using statistical inference of alternative splicing in lieu of direct evidence (Conesa et al, 2016; Shen et al, 2014).

To address these limitations and to directly detect transcript isoform changes, we performed Oxford nanopore (ONT) long-read sequencing of RNA obtained from distinct differentiation time points (Fig. 4A). The correlation between biological replicate measurements within a given time point was high (Spearman's correlation coefficient: >0.93), indicating the robustness of this measurement (Fig. EV3A,B). The correlation between time points was lower (Fig. EV3A,B), which was consistent with the observations from the RNA-seq data. The ONT-seq reads can span several exon-exon junctions (Fig. 4B), often including reads that

encompass the entire transcript, as exemplified for *MMP23B* (Fig. 4C). ONT-seq therefore provides a more direct view of expressed isoforms at different stages during neurogenesis. Although the fraction of reads that span one exon-exon junction was similar between both data types, ONT-seq clearly outperforms RNA-seq with regard to the fraction of reads that span at least two exon-exon junctions (Fig. 4B).

ONT-seq data correlated well with the RNA-seq data for the individual time points during neurogenesis (Fig. 4D). The correlation was highest for the iPS cell-state (day 0; >0.78) and lowest for day 5 of differentiation (>0.69). We observed a similar trend for the correlation between RNA isoforms detected by ONT-seq and RNA-seq for the different time points, although the correlations were lower as compared to gene-level expression (Fig. 4E). This similarity of RNA isoform expression captured by ONT- and RNA-seq was also observed at the level of single genes, such as for *PFN2* (Fig. 3B and Appendix Fig S2). Despite this correlation, we also observed clear differences in the detection of expressed RNA isoforms. For instance, in case of *MMP23B*, a gene that was turned on at an early stage during neurogenesis, RNA-seq inferred a different set of isoforms as compared to ONT-seq (Fig. EV3C). Furthermore, the detected expression levels of the different MMP23B isoforms strongly differed between the two data types (Fig. EV3C).

Together, these findings show that ONT-seq allows direct observation of transcript isoforms and their expression changes during human neuronal differentiation. Despite the correlation of detected RNA isoform changes between ONT- and RNA-seq, differences were detected on the single-gene level.

## Nanopore sequencing reveals new RNA isoforms at all stages of neurogenesis

Given the extended read length obtained by nanopore sequencing, on average 1096 nt compared to 100 nt for short-read Illumina sequencing, we wondered if the ONT long-read data will allow the identification of novel transcript isoforms that are expressed during the differentiation time course. To address this question, we first assembled a new transcriptome from nanopore reads. In the next step, we compared the new transcriptome to the human reference genome for assembly correction (Methods). We removed putative RNA isoforms that arose from artifacts during the RNA-seq library preparation, mainly from mispriming during reverse transcription and due to template switching (Verwilt et al, 2023). Furthermore, we excluded potential RNA isoforms that originated from closely spaced alternative transcription start sites (TSSs) or alternative

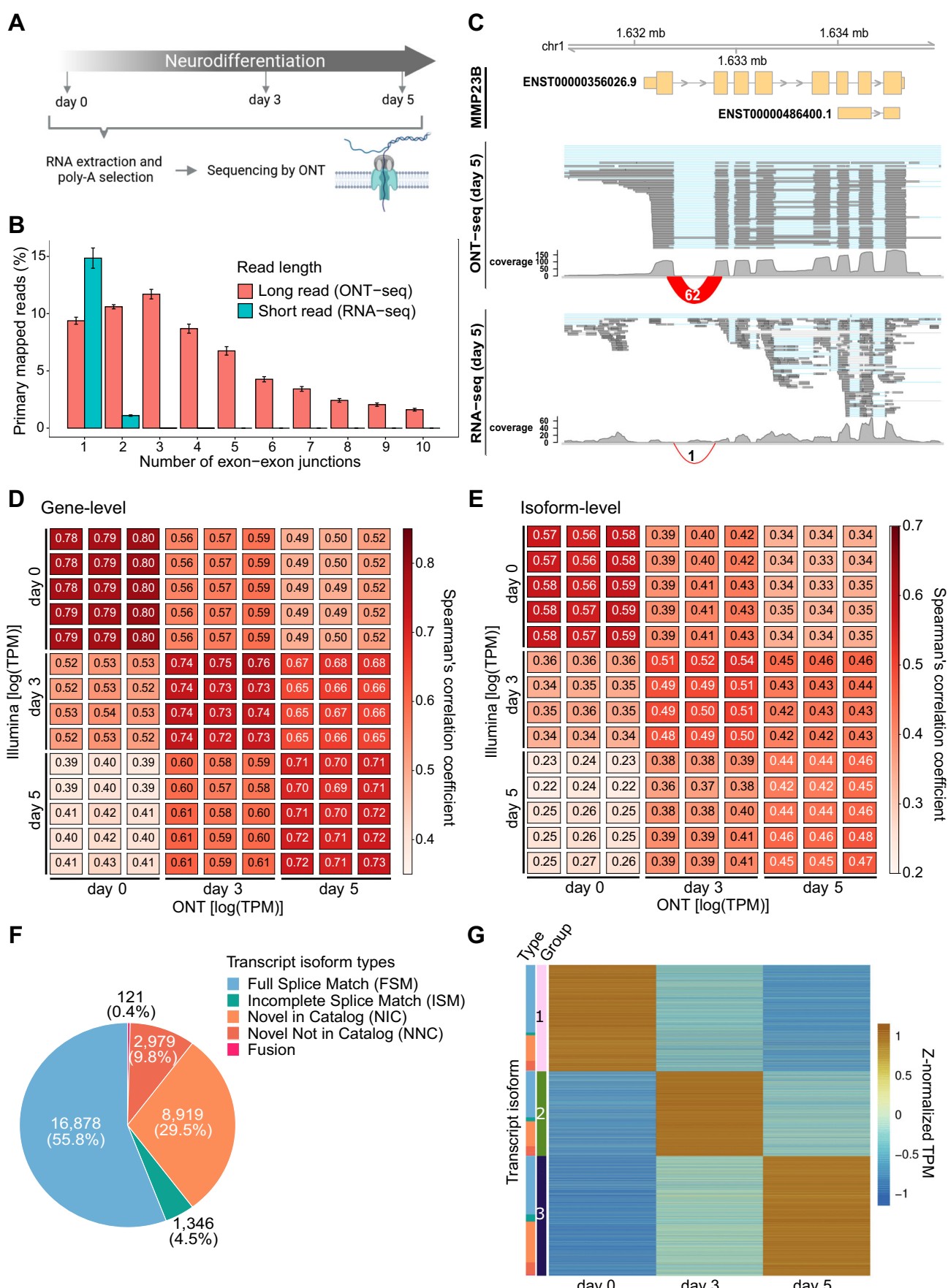

**Figure 4.  ONT long-read sequencing directly displays RNA isoform changes and uncovers new isoforms.**

(A) Scheme of the ONT-seq time-course experiment. (B) Proportions of primary mapped reads covering different numbers of exon-exon junctions within 18,724 multi-exon protein-coding genes in both ONT-seq reads and RNA-seq reads. Mean count of short reads spanning k junctions: 12,091,193 (k = 1), 896,729 (k = 2), 16,047 (k = 3), 387 (k = 4), 1 (k = 5), 0 (k = 6), 0 (k = 7), 0 (k = 8), 0 (k = 9), 0 (k = 10); mean count of long reads spanning k junctions: 635,381 (k = 1), 639,150 (k = 2), 664,531 (k = 3), 485,188 (k = 4), 371,051 (k = 5), 237,734 (k = 6), 189,300 (k = 7), 132,843 (k = 8), 110,992 (k = 9), 86,266 (k = 10). The error bar represents the standard error of the mean of the percentages. (C) Alignment of long reads from ONT-seq and short reads from RNA-seq obtained for day 5 to two transcript isoforms (ENST00000356026.9 and ENST00000486400.1) of *MMP23B*. The red arcs indicate the number of reads split across the junction between the first and second exons of ENST00000356026.9. (D) Spearman's correlation coefficient of gene expression levels (transcript per million, TPM) between replicates for each day of Illumina short-read RNA-seq and ONT-seq. (E) Spearman's correlation coefficient of transcript isoform expression levels (TPM), performed in the same manner as in (D). (F) Fractions of five main transcript isoform types identified in the ONT-seq time-course data. Transcript isoform types were classified as proposed in a recent study (Lienhard et al, 2023). A graphical illustration of transcript isoform types is given in Fig. EV3D. 'Catalog' refers to the set of annotated splice sites. (G) The heatmap shows z-normalized expression of transcript isoforms identified across three differentiation time points, with k-means clustering applied to group transcript isoforms. The color-coding for transcript isoform types matches that in (F). Number of transcript isoforms in Group 1: 10,410; Group 2: 8294; and Group 3: 11,701.

polyadenylation (pA) sites. This allowed the identification of 12,019 non-annotated transcript isoforms representing 40% of all detected RNA isoforms during neurogenesis (Figs. 4F and EV3D). 20% of the newly identified isoforms had at least one new splice site or a new combination of known splice sites, respectively (Fig. 4F). An interesting example was a newly discovered DHX15 isoform characterized by the retention of the intron between exon 10 and 11 of *DHX15* (Fig. EV3E), a gene encoding for an RNA helicase. In contrast to the known DHX15 isoform ENST00000336812.5, this novel isoform is predicted to lack the region encoding for the oligonucleotide/oligosaccharide binding (OB)-fold at its C-terminus (Appendix Fig. S3).

Thousands of new RNA isoforms were detected for all differentiation time points (Fig. EV3F). 3392 (28%) of novel transcript isoforms showed significant differential expression in at least one day-to-day comparison. Most new isoforms were observed at day 3 of neuronal differentiation (Fig. EV3F). Most frequently new isoforms arose from the use of alternative TSSs followed by exon skipping (single or multiple exons) and usage of alternative pA sites (Fig. EV3G). A subset of newly detected isoforms originated from readthrough transcription of RNA polymerase II leading to fusion transcripts of two genes (Fig. 4F). A large set of the new RNA isoforms (Group 1: 33%; Group 2: 40%; Group 3: 45%) are expressed in a differentiation stage-specific manner (Fig. 4G). The cluster analysis revealed three main clusters of RNA isoforms that differ in their timing of peak expression level during the differentiation course (Fig. 4G).

Together, ONT long-read sequencing revealed a large set of non-annotated RNA isoforms that are differentially expressed during human neural differentiation allowing a greater complexity of isoform changes.

## Dynamic transcript isoform changes remodel protein identity and function during neurogenesis

We next analyzed the potential functional implications of the widespread transcript isoform changes. To address this, we first performed a transcriptome-wide predictive analysis to determine whether RNA isoform changes can globally affect the protein encoding potential. This analysis revealed that alterations in the expression of isoforms can have a substantial impact on the amino acid sequence of the encoded protein. Main predicted consequences of detected transcript isoform changes were extension or shortening of the open reading frame or alterations in the domain

organization such as switch, gain or loss of a domain of the protein of interest (Fig. 5A).

Domain changes are of particular interest since they can directly alter protein functions. We therefore investigated the impact of transcript isoform changes on the protein encoding potential in more detail using the isoTV tool that we recently developed to visualize effects of RNA isoform changes on the protein sequence and post translational modification (Annaldasula et al, 2021). For instance, in case of *PFN2*, isoTV predicted that the change in the expression of the two main RNA isoforms PFN2a and PFN2b (Fig. 3B; Appendix Fig S2), alters the amino acid sequence in the C-terminal region of the Profilin domain (Fig. 5B). Of note, also the known phosphorylation site Ser130 is affected (Fig. 5B; (Walter et al, 2020)). The amino acid residues that differ in both PFN2 isoforms are enriched at the accessible surface area of PFN2 altering the interaction face (Fig. 5B) and likely also its function.

We next asked whether the observed changes in the expression of transcript isoforms can provoke real differences in the expression of protein isoforms during neurogenesis. To address this question, we performed a quantitative whole-cell mass spectrometry analysis at different time points during differentiation (Fig. 5C). This analysis revealed global dynamic proteome changes during the whole differentiation course (Fig. 5D,E). The proteomes at the endpoints of differentiation (day 0 and day 5) were most distinct (Fig. 5D,E) which was consistent with the observations for the dynamic transcriptome changes (Fig. 2C). The cluster analysis revealed four main groups of proteins that significantly changed during neural differentiation (Fig. 5E). Whereas the abundance of proteins in the first two clusters decreased, the relative levels of proteins in clusters three and four increased during neurogenesis and dominated the proteome at day 5. Data integration revealed that changes in protein levels correlated well with abundance changes of corresponding RNAs during neurogenesis (Fig. EV4A). This correlation further increased when only genes with significant RNA changes during the differentiation course were considered (Figs. 5F and EV4B). Notably, the correlation of protein expression changes was similarly high to RNA expression changes of the previous time point indicating that protein expression changes follow the observed RNA changes and persist longer, up to two days, during the differentiation course (Figs. 5F and EV4A,B).

We also used the proteome data to identify expressed protein isoforms based on detected isoform-specific peptides using a custom proteome that we built from the ONT-seq data. This approach allowed us to identify 132 genes with distinct changes in

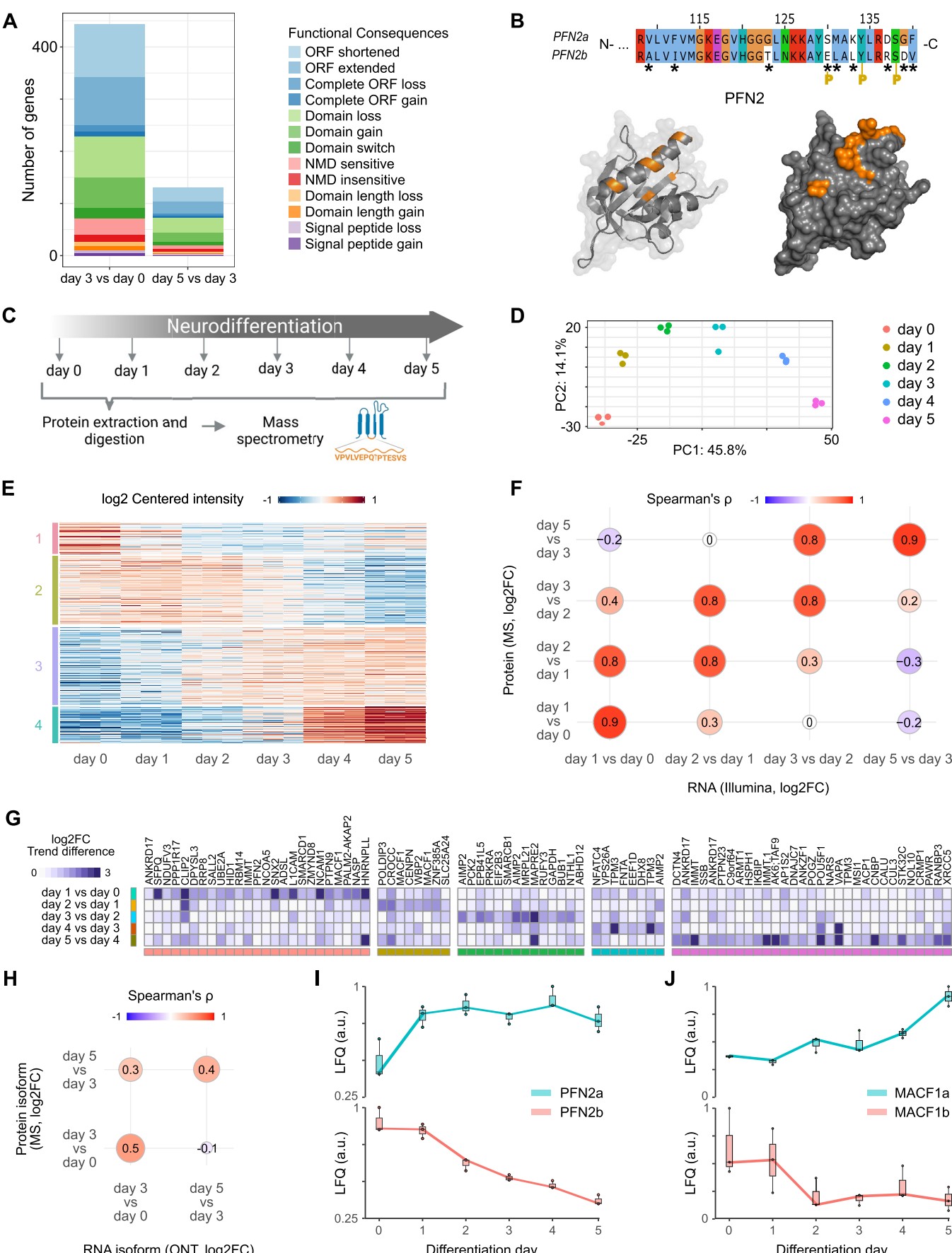

**Figure 5.  Functional consequences of dynamic transcript isoform changes.**

(A) Number of genes exhibiting isoform expression changes in ONT-seq time-course data with predicted functional consequences at the protein level when comparing day 3 to day 0 and day 5 to day 3. ORF: Open Reading Frame; NMD: Nonsense-mediated Decay. (B) Amino acid sequence of the C-terminal part of PFN2 that contains sequence differences between the two protein isoforms indicated by asterisks. P: phosphorylation sites (top). PFN2 structural models shown as band ribbon (left) and as surface representation (right). Residues that differ in PFN2 protein isoforms are highlighted in orange. Models were generated with Pymol version 2.5.5. using PDB file 1D1J (below). (C) Scheme of the whole-cell proteomics time-course experiment. (D) PCA plot of the top 500 most variable detected proteins throughout the differentiation time course as measured by mass spectrometry. (E) Heatmap representation of log2 centered normalized intensity of all proteins exhibiting significant abundance changes (padj <0.1 and |log2FC| > 0.32, empirical Bayes moderated t-test) during the differentiation course ($n = 1199$). We grouped the proteins into four clusters using k-means ($k = 4$). (F) Spearman's correlation coefficient of log2 fold changes of RNA and proteins between indicated days for differentially expressed genes (RNA: padj <0.05 and |log2FC| >0.59, Wald test; proteins: padj <0.1 and |log2FC| >0.32, moderated t-test). RNA was measured by RNA-seq and proteins by mass spectrometry (MS). Correlation analysis was restricted to genes and proteins that were differentially expressed between at least one time point. (G) Heatmap representation of the differences in the observed pairwise protein isoform changes (absolute difference of the logarithmic fold-changes ≥1, $n = 86$) from 76 genes upon the indicated days of neuronal differentiation. We grouped the proteins into five clusters using k-means ($k = 5$) and sorted them hierarchically. (H) Spearman's correlation coefficient of log2 fold changes from RNA and protein isoforms between indicated days. Log2 fold changes were measured by ONT-seq and mass spectrometry (MS). Correlation analysis was restricted to protein isoforms competing with at least one other protein isoform from the same gene ($n = 272$). (I, J) The boxplots show protein isoform quantifications (LFQ intensities, normalized to 1) across differentiation days for three biological replicate measurements. Protein isoform changes are shown for PFN2 (I) and MACF1 (J). The median values are depicted as the center. The box is defined by the first to the third interquartile range. The whiskers extend this interquartile range by a factor of 1.5, not exceeding the minimum or maximum values. Biological replicate measurements are depicted as individual dots.

the expression of protein isoforms between time points during human neuronal differentiation. Furthermore, this approach enables the identification of novel protein isoforms. For example, we identified a new expressed protein isoform for IMMT, a mitochondrial inner membrane protein. The corresponding RNA transcript (TCONS_00067861) encodes the new peptide sequence (ELDSITPEVLPGWKGMSD) detected and identified in our isoform-specific proteomics analysis. No NCBI entries are reported for the corresponding amino acid sequence to date.

For 76 proteins, representing 58% of the cases, we observed trend differences (absolute log2FC difference ≥1) in the protein isoform abundance during the differentiation course (Fig. 5G). Changes in the expression of protein isoforms occurred during all phases of neurogenesis (Fig. 5G) and correlated with RNA isoform changes as determined by ONT-seq and RNA-seq (Figs. 5H and EV4C; Spearman's correlation coefficient: 0.3–0.5). For instance, PFN2 underwent a protein isoform trend change during the differentiation time course (Fig. 5I). A main trend change in the abundance of the two known expressed PFN2 protein isoforms, PFN2a and PFN2b, occurred between day 0 and 1 (Figs. 5G,I and EV4D). A similar protein isoform trend change during early neurogenesis was detected for MACF1 (Figs. 5G,J and EV4E), a protein that is required to connect actin microfilaments with microtubules of the cytoskeleton (Cusseddu et al, 2021). In this case, we detected two non-annotated protein isoforms TCONS_00001949 and TCONS_00001963. For SMARCB1, a subunit of the BAF chromatin remodeling complex (Gourisankar et al, 2023), we detected a clear protein isoform trend change at a later phase during differentiation, between day 2 and 4 (Figs. 5G and EV4F). Mutations in *PFN2*, *MACF1,* and *SMARCB1* have been linked to neuronal diseases (Murk et al, 2021; Dobyns et al, 2018). Of note, the protein isoform trend change recapitulates the corresponding RNA isoform change as measured by RNA- and ONT-seq (Fig. EV4D–F). The overall number of identified protein isoform changes was likely an underestimate, since only a small fraction of peptides detected by MS originated from isoform-specific protein regions and were informative for isoform identification.

Together, this data indicates that dynamic changes in RNA isoform expression can lead to alterations in protein isoforms with likely different functions during human neurogenesis.

## Distinct sets of RBPs underlie widespread dynamic isoform changes during neurogenesis

We next analyzed potential determinants of widespread isoform changes during human neuronal cell differentiation. Given the prominent role of RNA binding proteins (RBPs) in alternative splicing (AS) regulation (Fu and Ares, 2014; Ule and Blencowe, 2019), we hypothesized that RBPs underlie the massive isoform shifts. To test this hypothesis, we first determined the expression profiles of RBPs during neurogenesis using our RNA-seq and ONT-seq time course data. This integrative analysis revealed that the expression of 1,204 RBPs strongly changed during the differentiation course (Fig. 6A). The cluster analysis identified four groups of expressed RBPs (Fig. 6A). RBPs of clusters 1 and 2 showed their peak expression level at the pluripotent state (day 0). On the contrary, the expression of RBP genes of clusters 3 and 4 peaked at day 3 or 5 of the differentiation course, respectively (Fig. 6A). This finding was method independent and indicates that different combinations of RBPs are present at distinct neurodevelopmental phases that may serve stage-specific functions during neurogenesis. A corresponding GO term analysis revealed terms for biological processes related to alternative splicing and splicing regulation that were significantly enriched for cluster 4 RBP genes (Fig. 6B). This observation suggests that RBPs of cluster 4 are implicated in alternative splicing regulation at later stages of neuronal differentiation when this group of RBPs is mainly expressed (Fig. 6A). In contrast, cluster 1 RBPs, associated with the pluripotent state, are strongly enriched in RBPs required for rRNA processing. For the other RBP expression clusters no significant GO terms for biological processes were obtained.

To unveil a more direct implication of RBPs in dynamic RNA isoform changes during neuronal differentiation, we next analyzed whether RNA binding motifs of sequence-specific RBPs were enriched in the vicinity of detected alternatively regulated exons. For this analysis we used rMAPS, a tool that performs an RBP motif search in the proximity of alternative exons (Park et al, 2016). This analysis uncovered binding motifs for a distinct set of RBPs that were significantly enriched within 250 nt of the alternatively skipped exons between different days of NGN3m differentiation

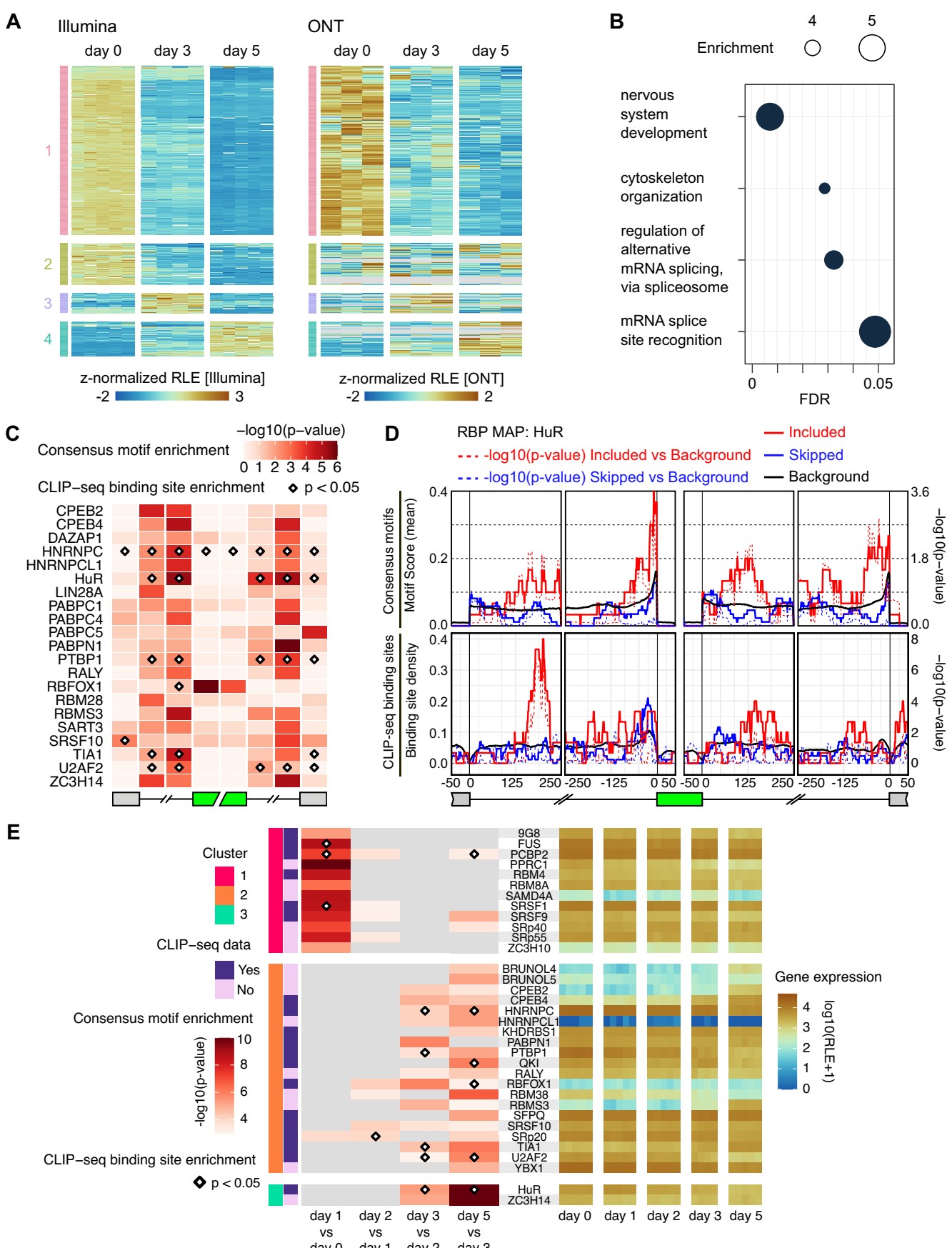

**Figure 6. Differential RBP expression and RNA binding correlate with isoform changes during neurogenesis.**

(A) The heatmap shows RLE (relative-log expression) and z-scaled gene expression data for genes associated with the GO terms RNA-binding (GO:0003723) and RNA splicing (GO:0008380). Depicted are genes with at least one differential expression event (Wald test, padj < 0.05; n = 1204). Expression values were measured by RNA-seq (Illumina; four to five replicates) and ONT-seq (triplicates). We grouped the proteins into four clusters using k-means (k = 4) and sorted them hierarchically. (B) Significantly enriched (FDR < 0.05) biological processes (GO slim) of cluster four shown in (A). Cluster one is enriched in proteins associated with rRNA processing (FDR = 0.02). Cluster two and three revealed no significant enrichments. (C) Heatmap illustrating the spatial enrichment patterns of RBP consensus motifs in the vicinity of differentially included exons between day 3 and day 2 of NGN3m neurogenesis. A schematic view of sequence segments scanned by rMAPS2 for the occurrences of consensus motifs in exon skipping events is shown below the heatmap. Alternatively regulated exons are depicted as a green box. 21 RBPs with a motif enrichment p-value (Wilcoxon's rank sum test) <0.001 are shown. Depicted is the smallest p-value within each region. The diamond-shaped labels indicate regions with enriched RBP binding sites (p-value < 0.05, Fisher's exact test) from CLIP-seq datasets (Data ref: Zhao et al, 2021). (D) The RBP MAP of HuR illustrates the spatial distribution of potential HuR binding sites in exon skipping events between day 3 and day 2 of the differentiation course. The same sequence segments as described in (C) were examined. Red indicates a higher inclusion level of the cassette exons in day 3 compared to day 2 (n = 30). Blue shows a lower inclusion level of the cassette exons (n = 86). Top: Histogram of consensus motifs of HuR generated by rMAPS2. Motif score, representing HuR motif density, is the percentage of nucleotides covered by the motif within a 50 bp sliding window. P-values were calculated by comparing motif scores between included versus background or skipped versus background exons using Wilcoxon's rank sum test. Below: Histogram of binding sites of HuR generated by RBP-Maps. Binding site density is the fraction of events that contain a binding site identified from CLIP-seq datasets (Data ref: Zhao et al, 2021) at each position. P-values were calculated by comparing binding site densities between included versus background or skipped versus background exons using Fisher's exact test. (E) Left: Heatmap depicting the temporal enrichment patterns of potential RBP binding sites relative to differentially included exons over the differentiation days. Plotted were 63 RBPs with enriched consensus motifs (p-value < 0.001, Wilcoxon rank sum test) in at least one region. For each RBP and day-to-day comparison, we showed the minimum enrichment p-value across different regions. RBPs were grouped using k-means (k = 4) clustering based on consensus motif enrichment p-values over the differentiation time. The diamond-shaped labels highlight the co-occurrence of enriched RBP binding sites (p-value < 0.05, Fisher's exact test) and enriched consensus motifs (p-value < 0.001, Wilcoxon's rank sum test) in the same region (Data ref: Zhao et al, 2021). Right: Quantification of total gene expression for RBPs from RNA-seq time-course data. Expression levels were normalized as relative-log expression (RLE) and log10 transformed.

(Figs. 6C and EV5A–C). Integration with available CLIP-seq data revealed a subset of predicted binding motifs in the vicinity of alternative exons that can indeed be bound by the corresponding RBP representing binding sites with higher confidence (Figs. 6C and EV5A–C).

Among the RBPs that were most significantly enriched at alternative exons between Day 2 and Day 3 of neurogenesis was the elav/Hu family protein HuR. The HuR binding motif was most strongly enriched in the regions 125 nt upstream and 250 nt downstream of alternatively skipped exons, and was associated with exon inclusion (Fig. 6D). Both regions in proximity to alternative exons are bound by HuR in cells as revealed by the integrative CLIP-seq analysis (Fig. 6C,D). HuR together with HNRNPC and TIA1 were among the RBPs with the highest motif enrichment scores that occupy upstream and/or downstream binding sites in proximity to alternative exons during Day 2 and 3 of neuronal differentiation, and that were strongly expressed (Fig. 6C,E). Elav/Hu family proteins have been implicated in neuronal differentiation and mediate neuronal RNA signatures by alternative splicing and alternative polyadenylation, and mutations have been associated with neurodevelopmental diseases (Hilgers, 2023). HNRNPC and TIA1 have also been implicated in AS regulation, also in the context of neurodevelopmental diseases (Niggl et al, 2023; Singh et al, 2011). Our findings suggest that these RBPs may cooperate in AS regulation during this differentiation phase.

The set of RBPs with significantly enriched binding motifs, a subset of which is indeed bound by the corresponding RBP, at alternatively regulated exons varied during neurogenesis (Figs. 6E and EV5D). These observations are consistent with the view that distinct sets of RBPs regulate AS changes at different stages during neurogenesis.

Taken together, these findings suggest that the coordinated expression of different combinations of sequence-specific RBPs at distinct differentiation phases can underlie the observed dynamic isoform changes during human neurogenesis.

## Discussion

Here, we established a model for the analysis of human neuronal cell differentiation in vitro that is as we show accessible for multi-omics analyses. In our systems-wide study, we provide evidence for widespread and dynamic changes in transcript abundance and identity that correlate with the massive cell morphology changes during neurogenesis. Nanopore sequencing directly uncovered thousands of new RNA isoforms, many of which are expressed in a differentiation stage-specific manner. Our work also sheds new light on the functional consequences and the determinants of isoform changes. This study illustrates the importance of post-transcriptional control mechanisms, primarily at the level of RNA processing dynamics, in the coordinated expression changes that underlie cell-state-transitions during human neurogenesis.

We found that the overall transcript levels and their alterations during human neuronal differentiation are often the sum of complex dynamic RNA isoform changes. This underlying layer of transcriptome changes during differentiation usually escapes detection by studies employing methods that lack isoform-resolution. Using nanopore sequencing that allows direct observation of transcript isoforms uncovered thousands of new RNA isoforms, whose expression can vary substantially during differentiation. These findings indicate an even greater complexity of transcriptome changes during human neuronal differentiation as originally anticipated. This view is consistent with pioneering work on RNA isoform changes during neurogenesis (Wu et al, 2010) and is also in line with previous observations in other differentiation models that have revealed a great variety in isoform expression (Fiszbein and Kornblihtt, 2017; Shi et al, 2014; Hu et al, 2013; Trapnell et al, 2010). An important future direction will be to elucidate which of the dynamic RNA isoform changes are functional. Our finding that a subset of them can lead to alterations in the expression of protein-isoforms at distinct stages of differentiation indicates that at least a set of RNA isoform changes can have functional consequences.

RNA isoform changes underlie main cell-state-transitions during human neuronal differentiation. The following findings support this view. First, we uncovered that RNA isoform changes are especially prevalent at early (between day 0 and 1) and late stages of the differentiation course (between day 3 and 5) coinciding with strong morphology changes. Second, we found that genes with dynamic transcript isoform changes are significantly enriched for processes involved in neuronal cell morphology changes including 'dendrite morphogenesis', 'axon guidance', 'synapse organization', and cytoskeleton-related processes. The latter finding was particularly interesting given the accumulating evidence that the actin and microtubule cytoskeleton plays a major role in neuronal morphogenesis including initiation, growth, guidance, and branching of axons and dendrites as well as in synapse formation (Luo, 2002; Compagnucci et al, 2016). Third, protein isoform changes were prevalent during early and late phases of neuronal differentiation, recapitulating the trends of corresponding RNA isoform changes. Consistent with the isoform changes, we detected major overall gene expression alterations, including transcription factors, at early and late stages of differentiation. These findings argue against a model of gradual transcriptome alterations during neuronal differentiation but rather suggest that phases of greater shifts alternate with phases of less changes.

Our data indicates that a substantial fraction of RNA isoforms (45%) during human neuronal differentiation originated from the use of alternative transcription start sites (TSSs) and alternative polyadenylation (pA) sites, rather than from alternative splicing (AS) sites. Although most previous studies have exclusively focused on AS as the main mechanism to generate RNA isoforms, alternative TSS and pA sites can impact the stability and translational efficiency of the RNA, and can result in protein isoforms. Our study is in line with previous work indicating that usage of alternative TSSs and pA sites contributes extensively to the diversification of transcriptomes in fully differentiated cells including neurons (Reyes and Huber, 2017; Shabalina et al, 2014; Pal et al, 2011; Furlanis et al, 2019). Far less is known about the importance, prevalence and dynamics of alternative TSS and pA site choice in diversifying the transcriptome during cell differentiation. Prior studies could reveal a lengthening of 3'-UTRs during murine neuronal cell differentiation due to the use of alternative pA sites (Hilgers et al, 2011; Miura et al, 2013). A more recent work provided evidence for widespread TSS switching during murine cerebellar development (Zhang et al, 2017). In line with these studies, our work is consistent with the view that differentiating cells, similarly to mature cells, rely on multiple mechanisms to dynamically generate RNA isoforms.

Distinct sets of expressed RBPs likely underlie dynamic and widespread changes in alternative splicing during human neurogenesis. We found that RNA binding motifs of a set of RBPs were enriched in proximity to AS sites suggesting that these RBPs serve a potential function in the corresponding AS event although more work is required to prove causality. Interestingly, the set of significantly enriched RBP motifs in the vicinity of alternatively regulated exons that can indeed be occupied by the respective RBP varied during neurogenesis supporting the view that distinct sets of RBPs underlie AS patterns at different phases during neuronal differentiation. Consistently, we found evidence for changes in the regulated expression of RBPs at different stages of neuronal

differentiation indicating that different combinations of RBPs are mainly present at distinct developmental times. This finding is in line with a previous proteome-wide study showing that a large set of RBPs undergo dynamic protein abundance changes during neuronal differentiation (Frese et al, 2017) and with dynamic RBP expression changes in other differentiation models (Wheeler et al, 2022; Zandhuis et al, 2021; Luo and Jiang, 2023). Our results are also consistent with the proposed model of 'alternative splicing regulatory networks' according to which specific sets of AS events are co-regulated by a subset of RNA sequence-specific RBPs (Ule and Blencowe, 2019). Dynamic alternative splicing regulatory networks have been detected during differentiation of various cell lineages including murine neuronal differentiation (Weyn-Vanhentenryck et al, 2018; Irimia et al, 2014; Baralle and Giudice, 2017). How the coordinated expression changes of RBPs are regulated and their specific role in dynamical changes of alternative splicing during neuronal differentiation are interesting future directions.

Current models of coordinated regulation of gene expression programs are primarily centered on the role of DNA-binding transcription factors. Our study now provides new evidence for RBPs and RNA processing, including alternative splicing, to constitute an additional post-transcriptional layer in the regulatory cascades that underlie neuronal differentiation. As a consequence, thousands of RNA isoforms are generated at distinct phases of neurogenesis that dynamically change during the differentiation course, a subset of them leading to similar isoform trend changes on the protein level. The importance of this regulatory layer is further illustrated by the observation that a dysfunction of RBPs and alterations in alternative splicing regulatory networks can cause neurodevelopmental disorders (preprint: Han et al, 2023; Lenzken et al, 2014; Parra and Johnston, 2022; Porter et al, 2018). The new human neuronal differentiation system that we have characterized in this study and our systems-wide approach that relies on integrative omics analyses can now be used to decode the post-transcriptional control layer underlying neurogenesis, and possibly also to illuminate how alterations contribute to neurodevelopmental disease.

## Methods

### Stem cell culture and cell differentiation

#### NGN3m cell line and cell culture
The NGN3m cell line was generated from the PGP1 (hu43860C) iPSC line that was reprogrammed using Sendai virus on adult dermal fibroblasts (Coriell GM23248) from Participant #1 of the Human Personal Genome Project (Coriell GM23338) (Ng et al, 2021; Church, 2005). The PGP1 iPSC line was modified to allow doxycycline-inducible expression of the neurogenin transcription factor Neurogenin-3 (NGN3). Briefly, iPSCs were transfected with the PBAN-NEUROG3 puromycin selectable piggyback vector (constructed from PB-TRE-dCas9-VPR (Addgene No. 63800)) to deliver NGN3 to the cells. Clones were selected with 1 µg/ml puromycin (Gibco, A1113803) as described previously (Sauter et al, 2019; Ng et al, 2021). Single-cell colonies were then picked and expanded to obtain the NGN3 monoclonal cell line, designated NGN3m (clone G12). NGN3m cells were cultured in feeder-free

maintenance medium for human embryonic stem (ES) and iPS cells mTeSR™1 (StemCell Technologies) containing 1% penicillin-streptomycin (Pen/Strep) on Matrigel® hESC-qualified matrix (Corning) coated plates, supplemented with 3.3 µg/ml Y-27632 RHO/ROCK pathway inhibitor (ROCKi) (StemCell Technologies) on the day of plating. Cells were cultured at 37 °C, 5% $CO_2$ and the medium was changed with fresh mTeSR™1 every 24 h. NGN3m cells were screened for chromosomal aberrations using the iPSC Genetic Analysis Kit (StemCell Technologies). Cells were regularly checked for mycoplasma.

### Induction of NGN3m differentiation

To induce neuronal differentiation of NGN3m, expression of the Neurogenin-3 transcription factor was initiated by addition of doxycycline (Sigma-Aldrich) to a final concentration of 0.5 µg/mL. In short, on the plating day 15,000–20,000 cells/cm² cells were plated onto Matrigel® coated plates in mTeSR™1 media with Y-27632 RHO/ROCK pathway inhibitor (ROCKi) (StemCell Technologies), to a final concentration of 3.3 µg/ml and grown for 24 h. The following day media was exchanged, and cells were grown in mTeSR™1 media. The second day after plating, doxycycline (0.5 µg/mL) was added to the culture media to induce Neurogenin-3 expression and cell differentiation. Media was changed every 24 h by mTeSR™1 with doxycycline for 5 consecutive days. Samples were collected every day in precise 24 h intervals: before the addition of doxycycline (D0) and every 24 h for 5 days (D1–D5). Cells were detached using Accutase™ (D0-D2) (StemCell Technologies), or incubation in 1 mL of DPBS (Gibco) with 5% FBS (Sigma-Aldrich) for 10 min (D3); 5 min (D4), or 0 min (D5). All cells were resuspended in 5 mL DPBS with 5% FBS, incubated for 10 min at RT and centrifuged at $200 \times g$ for 4 min at RT. The cell pellet was stored at −80 °C until further processed.

## RT-qPCR and RT-qPCR data analysis

Cells were grown in 6-well plates as described for induction of NGN3m differentiation. Cells were lysed on day 0, 1, 2, 3, 4, and day 5 of NGN3m differentiation in biological triplicates in 700 µL QIAzol™ and stored at −80 °C until further processed. RNA was isolated using the miRNeasy Micro Kit (Qiagen), including DNase treatment according to the manufacturer's instructions (Qiagen). The concentration of the isolated RNA was determined using the Qubit™ RNA HS Assay Kit (Thermo Fisher Scientific). Reverse transcription was performed for 1 µg of RNA (per sample) using the QuantiTect Reverse Transcription Kit (Qiagen) according to the manufacturer's instructions. cDNA was amplified using the PowerUp™ SYBR kit (Applied Biosystems). All primers that were used for the RT-qPCR analysis are listed in Appendix Table S1. All genes of interest were normalized to the geometric mean expression signal of the three human reference genes *FAM155B*, *SDE2*, and *ACTB*. Gene expression was calculated using the relative ΔΔCt method (Fleige and Pfaffl, 2006) as described recently (Taylor et al, 2019). Log2 fold changes of gene expression compared to day 0 were reported for each day during the differentiation time course. Horizontal bars represent the mean of the log2 transformed fold change ($n = 3$) and error bars represent the standard deviation of the log2 transformed fold change.

## RNA-sequencing (RNA-seq) analysis

### RNA-seq library preparation and sequencing

Cells were grown in 6 cm dishes and collected on day 0, 1, 2, 3, and day 5 of differentiation in five biological replicates as described for the induction of NGN3m differentiation. Cell pellets were lysed in 700 µL QIAzol™ and total RNA was isolated using the miRNeasy Micro Kit (Qiagen), including DNase treatment according to the manufacturer's instructions for RNA isolation, also of smaller RNA species >17 nt. (Qiagen). Five replicates for each day were used for library preparation. RNA-seq libraries were prepared using the KAPA RNA HyperPrep Kit with RiboErase (Roche) according to the manufacturer's instructions. No blinding was performed. The RNA-seq libraries were sequenced on a NovaSeq6000 system in paired-end mode (P-100). One of the Day 3 replicates appeared as an outlier (containing only 0.2% mapped reads) when processed and was excluded from further analysis.

## RNA-seq data analysis

### Gene and transcript expression quantification

The reference transcriptome, representing a collection of reference transcript isoforms, was constructed with the human reference genome GRCh38 version p12 and the corresponding GENCODE annotation version 28 using RSEM v.1.2.25 (Li and Dewey, 2011). RNA-seq reads were then aligned to these reference transcripts using Bowtie v.1.3.1 (Langmead et al, 2009). The resulting alignments were employed to calculate the expected read counts for each isoform using the Expectation-Maximization (EM) algorithm. The EM algorithm is designed to maximize the likelihood that a read originates from a particular isoform, given the current estimates of isoform expression. Of note, the process of assigning reads to isoforms is an approximation, as it relies on a statistical model to address uncertainty in read mapping when multiple isoforms share similar sequences, and it assumes that all possible isoforms in a transcriptome are known (Li et al, 2010).

### Comparative transcriptome analysis with data of the BrainSpan Atlas

Gene expression profiles obtained by RNA-seq were downloaded from the "Developmental Transcriptome" dataset within the BrainSpan Atlas of the Developing Human Brain (Data ref: Miller et al, 2014). This dataset encompassed RPKM values for 52,376 genes from 524 tissue samples, spanning a developmental timeline from 8 postconceptional weeks to 40 years of age, and from 26 distinct brain structures. To ensure consistency in data analysis, RPKM values were converted to TPM values. To minimize bias introduced by genes with negligible expression, genes with a mean TPM value of less than 1 in both the BrainSpan samples and our NGN3m cells were excluded from the analysis, resulting in a subset of 14,345 genes that were commonly expressed. To facilitate meaningful comparisons across different brain structures and developmental stages, we retained 16 brain structures with at least 24 time-point samples and 30 age categories with at least two structure samples.

We evaluated the correlation of gene expression levels between each BrainSpan sample and the day 5 NGN3m cells using Spearman's correlation coefficient.

### Differential gene expression analysis

Differential gene expression (DGE) analysis was performed using DESeq2 (Langmead, 2010; Love et al, 2014). The count matrix from the RSEM output and metadata including sample information were imported using the DESeqDataSetFromTximport function. Standard DGE analysis was performed through estimateSizeFactors, estimateDispersions, and nbinomWaldTest. We considered genes with an absolute log2 fold change (shrunk with the "ashr" option) >0.59 (gene expression increased or decreased at least 50%) and an FDR-adjusted $p$-value (Wald test) <0.05 to be differentially expressed in pairwise day-to-day comparisons.

Genes that were differentially expressed between any two of the time points were clustered into four groups based on their TPM values over time using the k-means function from the R stats package. The optimal k for k-means clustering was determined by the silhouette coefficient.

### Gene ontology (GO) enrichment analysis

Significant GO terms were obtained by PANTHER (Thomas et al, 2022), applying hypergeometric tests on the GO databases (full or slim) for biological processes. In the respective analyses, we selected GO terms with an FDR < 0.05 or <0.01. For hierarchically clustered groups of significant GO terms, we reported the most specific GO term at level 0.

### Differential transcript usage analysis

Identification of genes with differential transcript usage (DTU) was performed using the R package IsoformSwitchAnalyzeR v.1.18.0 (Vitting-Seerup and Sandelin, 2019). Transcript usage refers to the composition of expressed transcript isoforms within a gene. The DTU analysis compares the relative within-gene abundance of a transcript isoform between conditions, highlighting shifts in transcript usage patterns. The transcript-level counts and TPM abundances were imported using the importIsoformExpression function. The GENCODE annotation version 28 for GRCh38.p12 was also used as input for the analysis of alternative transcription and splicing. We only considered genes that showed sufficient expression at all time points (mean TPM > 1) and for which at least two transcript isoforms were detected. The DTU analysis was performed using isoformSwitchTestDEXSeq (Anders et al, 2012) function of IsoformSwitchAnalyzeR. We defined transcript isoforms with an FDR-corrected $p$-value (likelihood-ratio test) cutoff of 0.05 and a minimum absolute change in isoform usage (dIF) of 0.1 as differentially used between days. Genes that featured at least one differentially used isoform in consecutive day-to-day comparisons were used for GO enrichment analysis as described above.

### Alternative transcription and splicing analysis

Alternative transcription and splicing analysis were run in two phases. First, IsoformSwitchAnalyzeR performed splicing classification based on full-length transcript isoforms and provided a global view of the number of alternative transcription and splicing events between days. Second, rMATS performed event-level differential splicing analysis for individual genes with DTU identified in IsoformSwitchAnalyzeR.

In the first phase, we used the analyzeAlternativeSplicing and extractSplicingSummary functions of IsoformSwitchAnalyzeR to annotate the potential splicing events involved in the generation of each transcript isoform by comparing it to the hypothetical pre-mRNA containing all the exons within a gene and predict the gains or losses of alternative transcription and splicing events within the upregulated transcript isoforms between days, respectively. These events include alternative transcription start sites (ATSSs), alternative transcription termination sites (ATTSs), which in this case means alternative use of polyadenylation sites, alternative 3' splice sites (A3), alternative 5' splice sites (A5), single exon skipping (SES), multiple exon skipping (MES), mutually exclusive exons (MEE), and intron retention (IR).

In the second phase, we first mapped the RNA-seq reads to the human reference genome (GRCh38.p12 and GENCODE v28) using STAR v.2.7.9a (Dobin et al, 2013). The read alignment files were then used by rMATS v.4.1.2 (Shen et al, 2014) to detect differential alternative splicing events between days. These events include skipped exons (SE), alternative 5' splice sites (A5SS), alternative 3' splice sites (A3SS), mutually exclusive exons (MXE), or retained introns (RI). The results based on counting junction reads only were selected and filtered such that all significant differential splicing events (FDR < 0.05, likelihood-ratio test) were detected within genes exhibiting a significant DTU.

## RNA binding protein (RBP) consensus motif enrichment analysis

Differential exon skipping events detected in rMATS were divided into differential inclusion (IncLevelDifference >0.05 and FDR < 0.05), differential exclusion (IncLevelDifference <−0.05 and FDR < 0.05), and unregulated background (FDR > 0.5). The genomic coordinates of these events were submitted to the rMAPS2 v.2.2.0 web server (Hwang et al, 2020) to calculate the motif scores of 91 default RBPs in the vicinity of the regulated exons under the default parameters. rMAPS2 scanned the following sequence segments for consensus motifs in exon skipping events: the first 50 bp of the 3'-end of the upstream exon, the first 250 bp of the 5'-end of the upstream flanking intron, the first 250 bp of the 3'-end of the upstream flanking intron, the first 50 bp of the 5'-end of the target alternative exon, the first 50 bp of the 3'-end of the target alternative exon, the first 250 bp of the 5'-end of the downstream flanking intron, the first 250 bp of the 3'-end of the downstream flanking intron, and the first 50 bp of the 5'-end of the downstream exon. To identify specific positions with significant differences in motif scores between regulated and background exons, the Wilcoxon's rank sum test was performed, and the smallest $p$-value within a region was used to represent the significance level for motif enrichment.

## RBP binding site enrichment analysis

We obtained the human RBP binding sites (build hg38) from the CLIPdb module of POSTAR3 (Zhao et al, 2021), which is a curated collection of CLIP-seq datasets from diverse technologies (i.e., HITS-CLIP, PAR-CLIP, iCLIP, eCLIP, etc.). Of the 91 RBPs analyzed for consensus motifs in rMAPS2, 36 were found to have binding sites available from CLIP-seq data. RBP-Maps v.0.1.4 (Yee et al, 2019) was used in peak-based mode to visualize the density of RBP binding sites in proximity to the regulated exons and to test whether the fraction of events with a binding site was significantly altered relative to the background using the Fisher's exact test (--sigtest fisher). The sequence segments to be scanned were consistent with those in the consensus motif enrichment analysis by

setting --exon_offset 50 and --intro_offset 250 and employing the identical sets of differentially regulated and unregulated background exons between days.

## Oxford Nanopore sequencing (ONT-seq) analysis

### ONT-seq library preparation and sequencing
The library was prepared using the direct cDNA sequencing (SQK-DCS109) kit from Oxford Nanopore Technologies according to the manufacturer's protocol. No blinding was performed. The sequencing was performed on the MinION sequencer for 48 to 72 h with −180 mV starting voltage using the R9.4.1 FLO-MIN106 flow cells.

## ONT-seq data analysis

### Transcript isoform detection and quantification
ONT-seq data was processed using the IsoTV pipeline (Annaldasula et al, 2021). In short, the data were basecalled using Guppy v.3.2.4 (Oxford Nanopore Technologies) with parameters corresponding to the used flowcell and kit. Low quality reads (mean Q < 5) were filtered out using Filtlong (Wick, 2023) (https://github.com/rrwick/Filtlong/). Then the full-length transcripts were identified and oriented using Pychopper (Oxford Nanopore Technologies, 2020a) (https://github.com/nanoporetech/pychopper) with the options "-x", "-b", "-S", "-a", "-u", and "-l". To de novo assemble the transcriptome, we applied Pinfish (Oxford Nanopore Technologies, 2020b) (https://github.com/nanoporetech/pinfish) to full-length transcripts from all samples with the following parameters: cluster_gff used "-c = 3", "-d = 5", "-e = 50", and "-p", collapse_partials used "-d = 5", "-e = 50", and "-f = 500", and polish_clusters used "-a" and "-c = 1". Then, the clustered and polished transcripts were compared to the human reference genome annotation, Gencode Version 32, using GffCompare (Pertea and Pertea, 2020) with the following parameters: "-r", "-R", "-A", and "-K". To obtain the transcript expression, all the reads with Q > 10 were aligned to the reference-corrected transcriptome using Minimap2 (Li, 2018) with the following parameter "-uf" (primary mapping rates: mean 0.729, min. 0.597, and max. 0.865). TPM values were calculated using read count values normalized with DESeq2 (Love et al, 2014), counting only primary mappings of the reads.

To better characterize the identified transcript isoforms and to remove potential artifact reads, we applied SQANTI3 v.5.1.1 (Pardo-Palacios et al, 2024) for quality control and curation. In the quality control phase, in addition to our constructed transcriptome from ONT-seq and GENCODE annotation version 32 as a reference, we incorporated cap analysis of gene expression (CAGE) peak data (human.refTSS_v3.1.hg38.bed) (Data ref: Abugessaisa et al, 2019; Takahashi et al, 2011), polyA site data (atlas.clusters.2.0.GRCh38.96.bed) (Date ref: Herrmann et al, 2019), and polyA motif data (mouse_and_human.polyA_motif.txt), provided by SQANTI3, and splice junction coverage data from our RNA-seq datasets. We labeled transcript isoforms arising from potential mispriming due to a misalignment of the reverse transcription (RT) primer during the RT step if there was a nucleotide sequence of at least 60% adenines following the detected pA site. Besides, SQANTI3 identified non-canonical splice junctions potentially caused by reverse transcriptase template switching

during RT. GeneMarkS-T was used to predict the protein-coding regions in RNA transcripts (Tardaguila et al, 2018).

The identified transcript isoforms were categorized into five primary classes similarly as proposed in recent studies (Lienhard et al, 2023; Tardaguila et al, 2018): full splice match (FSM, transcripts matching a reference transcript at all splice junctions), incomplete splice match (ISM, transcripts matching consecutive but not all splice junctions of the reference transcripts), novel in catalog (NIC, transcripts containing new combinations of annotated donor or acceptor sites), novel not in catalog (NNC, transcripts using novel donors and/or acceptors), and fusion (transcript spanning two annotated loci).

In the curation phase, we defined specific filtering rules tailored to each isoform type. FSM and ISM transcripts were retained if (1) they were not potential RT mispriming products; (2) none of their splice junctions were a potential RT-template switching artifact; (3) their putative TSS fell within the reference TSS; (4) their putative TTS fell within the reference polyA site; and (5) a polyA motif was detected close to the putative TTS For NIC, NNC, and fusion transcripts, they were kept if they satisfied the first two criteria mentioned above, and all splice junctions were either canonical or supported by a minimum of 10 reads.

### Differential transcript usage analysis
Differential transcript usage (DTU) analysis followed a similar procedure as described in the RNA-seq analysis section. The main difference was that we provided the IsoformSwitchAnalyzeR program with the curated transcriptome from the ONT-seq data. In addition to the analysis of alternative transcription and splicing, we used the analyzeSwitchConsequences function of IsoformSwitchAnalyzeR to predict the functional consequences of the upregulated transcript isoforms between days. Pfam Scan v.1.6 (Madeira et al, 2022) for protein domain search and SignalP v.5.0 (Armenteros et al, 2019) for signal peptide prediction were applied, following the default instructions. Nonsense-mediated decay (NMD) sensitivity was inferred based on the presence of premature termination codons (PTCs) located at least 50 bp upstream of the last exon-exon junction. Open reading frame (ORF) sequence similarity was determined by comparing amino acid sequences translated from the transcript nucleotide sequences. Functional features of the transcript isoforms and their predicted translated products were visualized by IsoTV (Annaldasula et al, 2021).

### Differential transcript expression analysis
Differential transcript expression (DTE) analysis was performed using the established workflow for DGE in DESeq2 as it also has demonstrated efficacy at the transcript level (Love et al, 2018). The main difference was that we provided the transcript-level count matrix to DESeq2. We considered transcripts with an absolute log2 fold change (shrunk with the "ashr" option) >0.59 (transcript expression increased or decreased at least 50%) and an FDR-adjusted $p$-value (Wald test) <0.05 to be differentially expressed in pairwise day-to-day comparisons.

## Quantitative Proteomics

Proteomics sample preparation was performed as previously described with minor modifications (Kulak et al, 2014). In brief,

0.5 million cells in three biological replicates for day 0, 1, 2, 3, 4, and 5 of differentiation were lysed under denaturing conditions in 300 μl of denaturing buffer (3 M guanidinium chloride (GdmCl), 10 mM tris(2-carboxyethyl)phosphine, 40 mM chloroacetamide, 100 mM Tris-HCl pH 8.5). Lysates were denatured at 95 °C for 10 min while shaking at 1000 rpm in a thermal shaker and sonicated in a water bath for 10 min. 30 μg protein (according to BCA) per sample was diluted with the dilution buffer (10% acetonitrile and 25 mM Tris-HCl, pH 8.0), to reach a 1 M GdmCl concentration. Then, proteins were digested with LysC (Roche, Basel, Switzerland; enzyme to protein ratio 1:50, MS-grade) while shaking at 700 rpm at 37 °C for 2 h. The digestion mixture was again diluted with the dilution buffer to reach 0.5 M GdmCl, followed by a tryptic digestion (Roche, enzyme to protein ratio 1:50, MS-grade) and incubation at 37 °C overnight in a thermal shaker at 700 rpm. Peptide desalting was performed according to the manufacturer's instructions (Pierce C18 Tips, Thermo Scientific, Waltham, MA). Desalted peptides were reconstituted in 0.1% formic acid in water and further separated into four fractions by strong cation exchange chromatography (SCX, 3 M Purification, Meriden, CT). Eluates were first dried in a SpeedVac, then dissolved in 5% acetonitrile and 2% formic acid in water, briefly vortexed, and sonicated in a water bath for 30 s prior to injection to nano-LC-MS/MS. LC-MS/MS Instrument Settings for Shotgun Proteome Profiling and Data Analysis LC-MS/MS was carried out by nanoflow reverse-phase liquid chromatography (Dionex Ultimate 3000, Thermo Scientific) coupled online to a Q-Exactive HF Orbitrap mass spectrometer (Thermo Scientific), as reported previously (Ni et al, 2019). No blinding was performed.

### Proteomics data analysis

Raw MS data were processed with MaxQuant software (v2.0.1.0) and searched against the human proteome database UniProtKB, including isoforms, with 78,120 entries, released in March 2021. The MaxQuant processed output files are accessible on PRIDE (PXD046084), showing peptide and protein identification, accession numbers, % sequence coverage of the protein, q-values, and label-free quantification (LFQ) intensities.

#### Differential protein enrichment analysis

Identified proteins were analyzed using the R package DEP (Zhang et al, 2018). In short, data was filtered for potential contaminants. Proteins were considered as hits if they were present in two or three out of three replicates in at least one condition. Missing values were imputed using the left-censored imputation method suitable for MNAR (missing not at random) assuming that the values that are missing originate from low abundance proteins under the detection limit of the instrument. To impute missing values, random samples were taken from a manually defined left-shifted Gaussian distribution with a shift of 1.8 and a scale of 0.3. Differential enrichment analysis was based on linear models and the empirical Bayesian approach as provided in the DEP package. The differential expression analysis was performed using the test_diff function. Proteins with an absolute log2 fold change >0.32 (increased or decreased protein abundance of more than 25%) and FDR-adjusted $p$-value (empirical Bayes moderated t-test based on the limma package (Ritchie et al, 2015) < 0.1 were considered as differentially enriched.

#### Detecting protein isoform changes

We processed the raw MS data with the MaxQuant software using our de novo custom proteome tailored to neuronal differentiation derived from the ONT-seq transcriptome by GeneMarkS-T (Tang et al, 2015). We obtained the unique peptide assignments and their measured intensities for all samples. Using FlashLFQ (Millikin et al, 2018) via Galaxy v1.0.3.0 (Afgan et al, 2022) with the '--mbr' parameter, we obtained normalized intensity values and protein isoform changes considering unique peptides exclusively. Finally, we sum the fractionated intensity values and gain protein isoform-specific LFQ values for each sample.

Direct comparisons between different proteins or their isoforms are generally invalid as peptides are not equally detectable, leading to incomparable signal intensity values. Instead, we examined the difference between the logarithmic fold changes occurring across differentiation days referred to as trend differences.

## Data availability

The datasets and computer code produced in this study are available in the following databases: Long- and short-read RNA-seq data: Gene Expression Omnibus GSE245325. Protein mass spectrometry data: PRIDE PXD046084. Computational methods and custom scripts: GitHub (https://github.molgen.mpg.de/MayerGroup/NGN3_paper_code).

The source data of this paper are collected in the following database record: biostudies:S-SCDT-10_1038-S44320-024-00039-4.

## Peer review information

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

## Acknowledgements

We thank Elisabeth Altendorfer and Geno Villafano for critical comments on the manuscript. We thank the Sequencing facility of the MPIMG for sequencing. We thank the Mass spectrometry facility of the MPIMG for proteomics measurements. This work was funded by the Max Planck Society (to AM). JU was supported by a doctoral scholarship of the Studienstiftung des deutschen Volkes, OJ by a 2017 FEBS Long-Term Fellowship and SA by a Fulbright Fellowship. VB acknowledges support by the Volkswagen Foundation

(Freigeist—A110720) and the Deutsche Forschungsgemeinschaft (EXC-2151-390873048-Cluster of Excellence—ImmunoSensation2 at the University of Bonn). Schematic views in Fig. 1A, Fig. 2A, Fig. 4A, and Fig. 5C were created with Biorender.com.

## Author contributions

**Jelena Ulicevic**: Software; Formal analysis; Validation; Investigation; Visualization; Methodology; Adapted and characterized the NGN3m cell line, analyzed the proteomics data. Helped with the analysis of RNA-seq data. **Zhihao Shao**: Data curation; Software; Formal analysis; Validation; Investigation; Visualization; Methodology; Analyzed RNA- and ONT-seq data. Re-analyzed the CLIP-seq data. **Olga Jasnovidova**: Data curation; Formal analysis; Validation; Investigation; Methodology; Performed RNA-seq, ONT-seq and proteomics experiments. **Annkatrin Bressin**: Data curation; Software; Formal analysis; Visualization; Performed protein isoform data and differential RBP expression analysis. **Martyna Gajos**: Data curation; Software; Formal analysis; Helped with the RNA- and ONT-seq data analysis. **Alex HM Ng**: Methodology; NGN3 cell line generation. **Siddharth Annaldasula**: Formal analysis; Helped with the proteomics data analysis. **David Meierhofer**: Methodology; Helped with the proteomics data analysis. **George M Church**: Resources; NGN3 cell line generation. **Volker Busskamp**: Supervision; Methodology; NGN3 cell line generation. **Andreas Mayer**: Conceptualization; Resources; Supervision; Funding acquisition; Writing—original draft; Project administration; Writing—review and editing; Planned and designed experiments, and had overall responsibility over the study.

Source data underlying figure panels in this paper may have individual authorship assigned. Where available, figure panel/source data authorship is listed in the following database record: biostudies:S-SCDT-10_1038-S44320-024-00039-4.

## Funding

## Disclosure and competing interests statement

The authors declare no competing interests.

# Expanded View Figures

**Figure EV1.   Characterization of NGN3m neuron-like cells (related to Fig. 2).**

(**A**) Z-normalized expression (transcript per million, TPM) of genes ($n = 103$) that show differential expression in at least one pairwise day-to-day comparison and that are included in the gene set 'KEGG_CELL_CYCLE' of the Molecular Signatures Database (Liberzon et al, 2015) during five days of NGN3m differentiation. The dashed vertical line indicates the transition into the post-mitotic phase. Cluster 1 ($n = 83$) and 2 ($n = 20$) include active genes in the mitotic phase (from day 0 to day 2) and the post-mitotic phase (from day 3 to day 5), respectively. (**B**) Z-normalized expression (TPM) of differentially expressed transcription factors (TFs, $n = 1094$) during five days of NGN3m differentiation. Cluster 1 ($n = 447$), 2 ($n = 268$), and 3 ($n = 379$) include TFs that were primarily expressed on day 0, from day 1 to day 3, and on day 5, respectively. (**C**) Z-normalized expression (TPM) of marker genes for different types of neurons. The neural marker genes are based on the "Neural markers guide" published by Abcam (Data ref: Abcam). All markers rely on experimental evidence. (**D**) The heatmap shows Z-normalized Spearman's correlation coefficients between day 5 NGN3m cells and the human developmental transcriptome data collection of the Developing Human Brain Atlas (Miller et al, 2014). The x-axis shows the developmental time, and the y-axis represents the brain regions. pcw: post-conception weeks; mos: months; yrs: years; MD: mediodorsal nucleus of thalamus; CBC: cerebellar cortex; S1C: primary somatosensory cortex (area S1, areas 3,1,2); IPC: posteroventral (inferior) parietal cortex; M1C: primary motor cortex (area M1, area 4); STR: striatum; V1C: primary visual cortex (striate cortex, area V1/17); A1C: primary auditory cortex (core); VFC: ventrolateral prefrontal cortex; HIP: hippocampus (hippocampal formation); ITC: inferolateral temporal cortex (area TEv, area 20); OFC: orbital frontal cortex; DFC: dorsolateral prefrontal cortex; MFC: anterior (rostral) cingulate (medial prefrontal) cortex; STC: posterior (caudal) superior temporal cortex (area 22c); and AMY: amygdaloid complex.

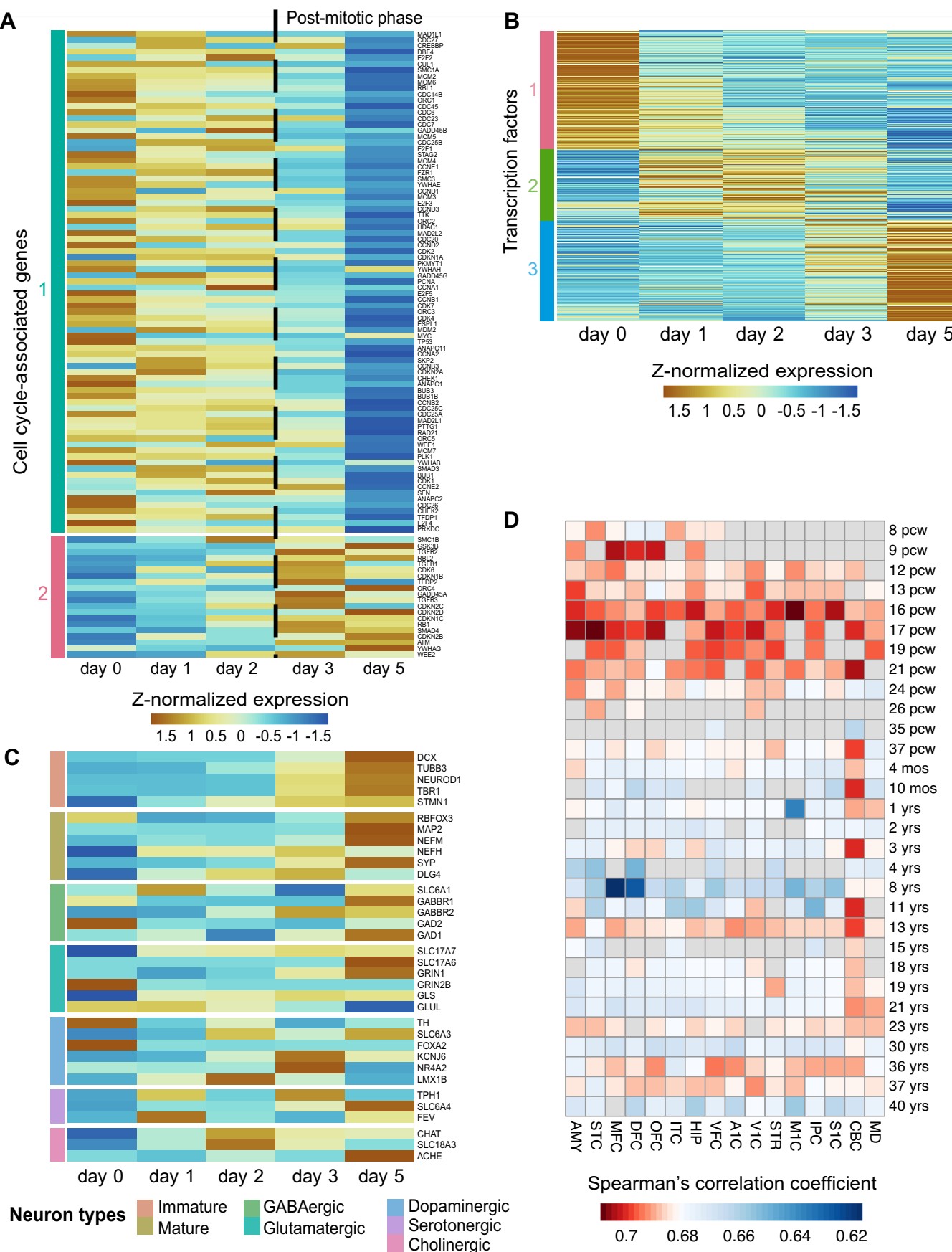

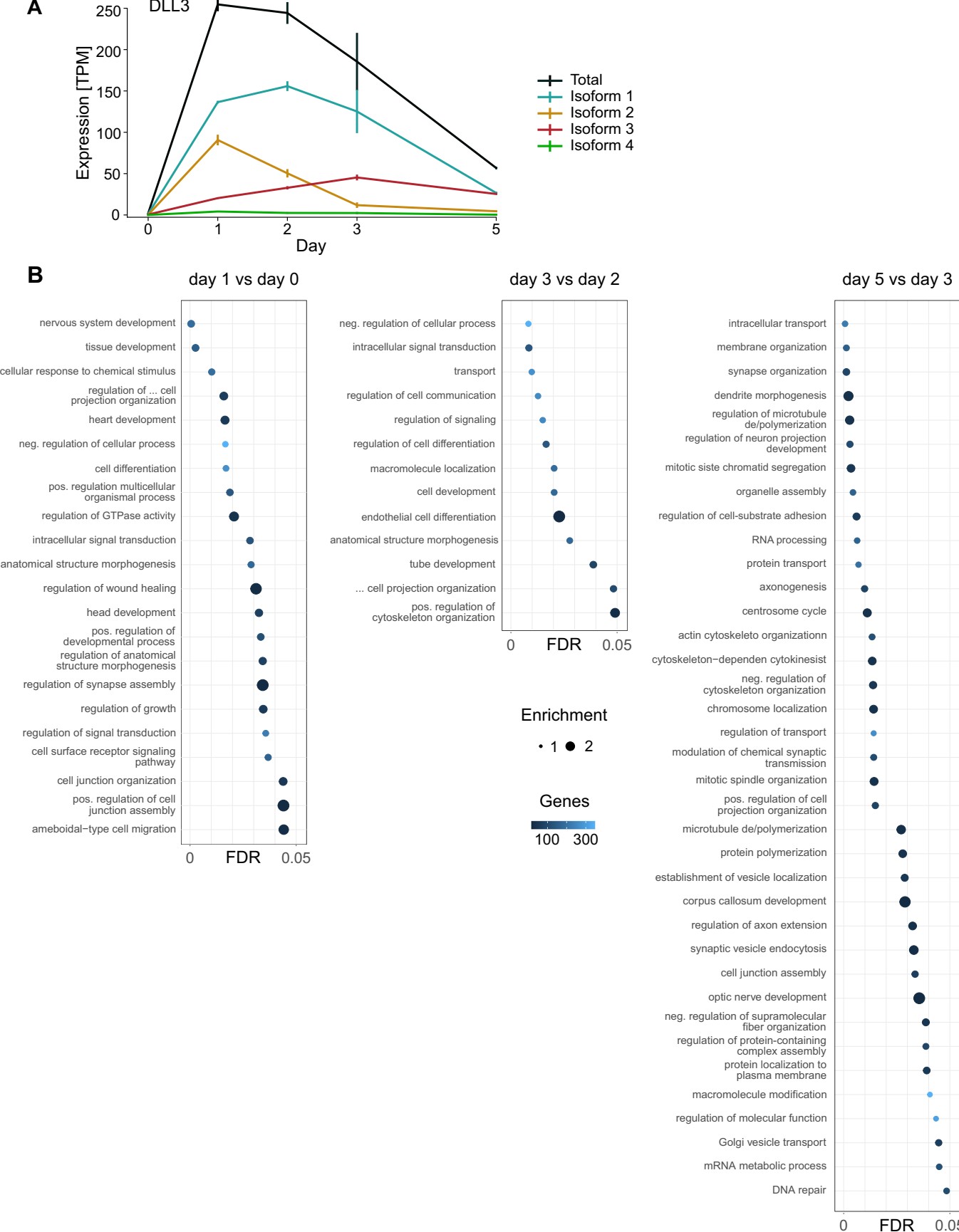

◀  **Figure EV2.   Significantly enriched gene ontology (GO) terms for genes with differential isoform expression patterns during NGN3m differentiation (related to Fig. 3).**

(A) Quantification of total gene expression of *DLL3* and relative abundance of four transcript isoforms inferred from RNA-seq time-course data. Expression levels were normalized as transcript per million (TPM). (B) All significant GO terms (Fisher's exact test, FDR < 0.05, level 0) from genes with differential isoform expressions between indicated days of the differentiation course.

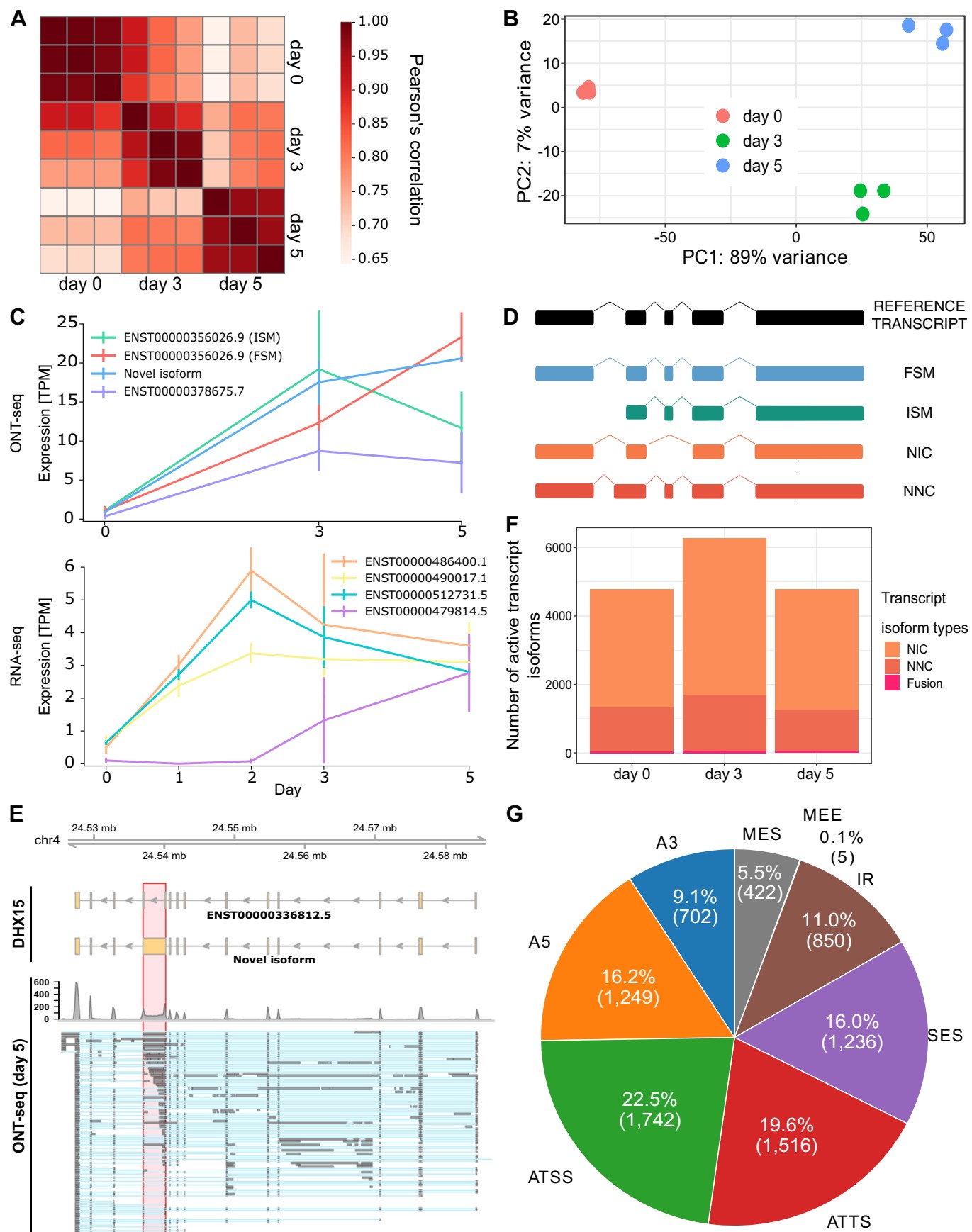

◀ **Figure EV3.  High reproducibility of ONT-seq replicate datasets and identification of new RNA isoforms during NGN3m differentiation (related to Fig. 4).**

(A) Pearson's correlation analysis of gene expression levels (transcript per million, TPM) of ONT-seq replicate datasets for the different time points. (B) PCA plot of expressed genes throughout the differentiation course as measured by ONT-seq. (C) RNA isoform expression profiles for *MMP23B* directly observed by ONT-seq (top panel) or inferred from RNA-seq data (lower panel). TPM: transcripts per kilobase million. ISM: Incomplete Splice Match. FSM: Full Splice Match. (D) Schematic view of main types of transcript isoforms that were identified from ONT-seq data and classified based on their agreement with the intron structure of the reference transcript by SQANTI3. FSM: matches all splice junctions (SJs) perfectly; ISM: matches the reference SJs partially; NIC: novel isoform with a new combination of known splice sites; and NNC: novel isoform with at least a new splicing site. (E) ONT-seq identifies a novel RNA isoform for *DHX15* that is expressed at day 5 of NGN3m differentiation. A red box highlights the differing region. (F) Number of new isoforms that show active expression (mean TPM > 5 across replicates) at day 0 (NIC $n = 3456$, NNC $n = 1275$, Fusion $n = 44$), day 3 (NIC $n = 4584$, NNC $n = 1617$, Fusion $n = 66$), and day 5 (NIC $n = 3518$, NNC $n = 1219$, Fusion $n = 53$) of NGN3m differentiation as measured by ONT-seq. (G) Fractions of alternative transcription and splicing mechanisms of differentially expressed isoforms, as measured by ONT-seq, across all consecutive day-to-day comparisons.

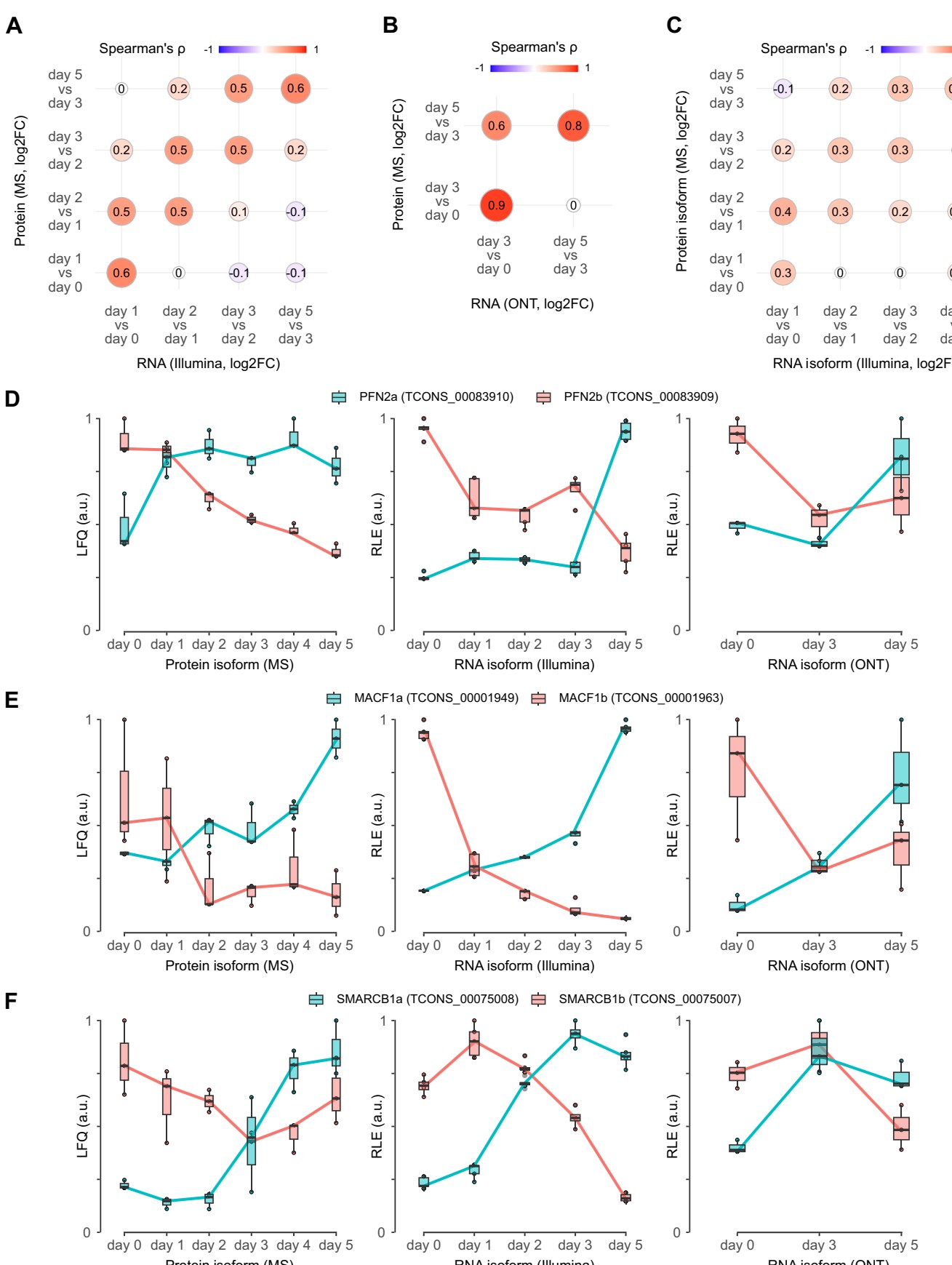

**Figure EV4.   Correlation of proteomics and transcriptomics data (related to Fig. 5).**

(A) Spearman's correlation coefficient of log2 fold changes of RNA- and protein-levels between indicated days for all expressed genes. RNA levels were measured by RNA-seq and proteins by mass spectrometry (MS). Correlation analysis was restricted to genes and proteins that were detected in at least one time point. (B) Spearman's correlation coefficient of log2 fold changes of RNA- and protein-levels between indicated days for differentially expressed genes (RNA: padj <0.05 and |log2FC| > 0.59, Wald test; proteins: padj <0.1 and |log2FC| > 0.32, moderated t-test). RNA levels were measured by ONT-seq and proteins by mass spectrometry (MS). Correlation analysis was restricted to genes and proteins that were differentially expressed between at least one time point. (C) Spearman's correlation coefficient of log2 fold changes of RNA and protein isoforms across consecutive days. Logarithmic fold changes were measured by RNA-seq and mass spectrometry (MS). Correlation analysis was restricted to protein isoforms competing with at least one other protein isoform from the same gene ($n = 272$). (D–F) Depicted are protein and RNA isoform quantifications for PFN2 (D), MACF1 (E), and SMARCB1 (F) across the differentiation days. The boxplots show protein isoform quantifications (LFQ intensities, normalized to 1) from mass spectrometry (MS, triplicates) (left) and RNA quantification (relative log expression (RLE), normalized to 1) from Illumina (four to five replicates) and ONT (triplicates) sequencing data (right). The median values are depicted as the center. The box is defined by the first to the third interquartile range. The whiskers extend this interquartile range by a factor of 1.5, not exceeding the minimum or maximum values. Biological replicate measurements are depicted as individual dots.

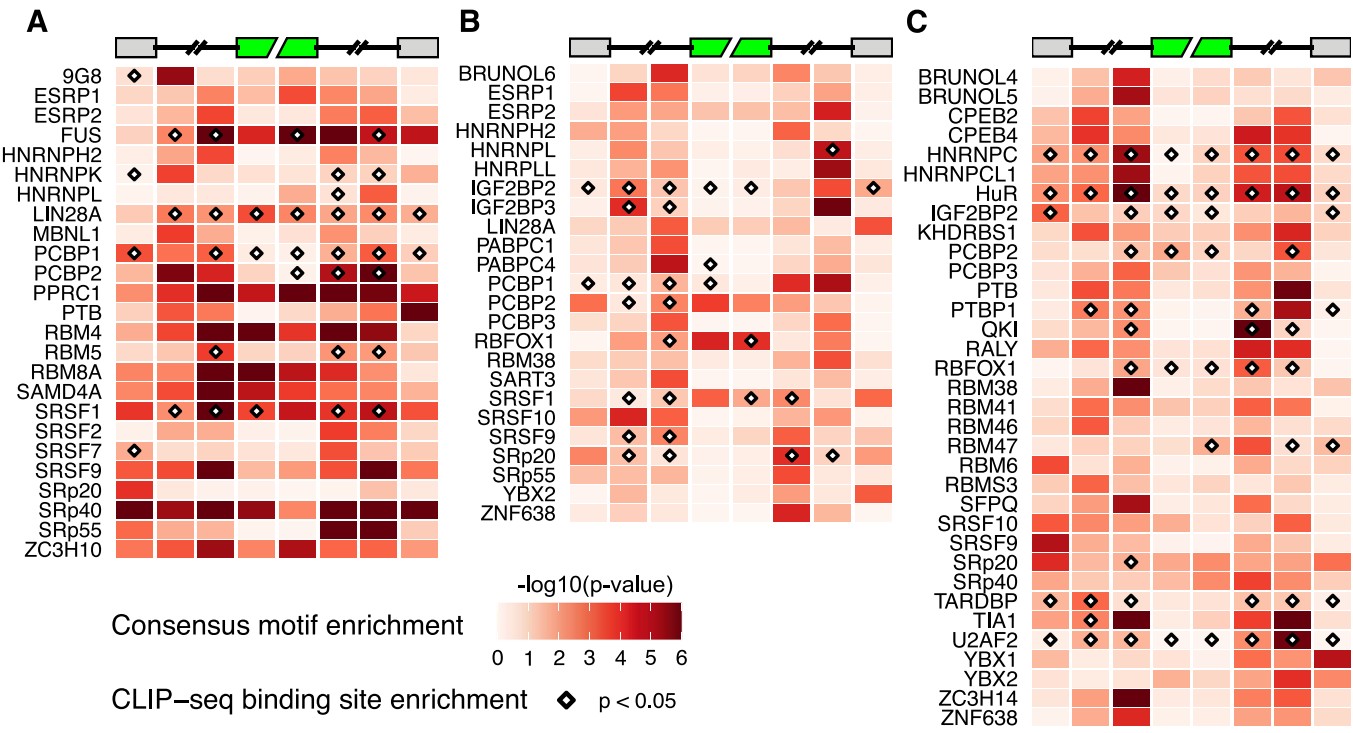

**Figure EV5.   Differential RNA binding patterns during NGN3m differentiation (related to Fig. 6).**

(A–C) Heatmap illustrating the spatial enrichment patterns of RBP consensus motifs in the vicinity of differentially included exons between (**A**) day 0 and day 1, (**B**) day 1 and day 2, and (**C**) day 3 and day 5 of NGN3m neurogenesis as determined by scanning motifs in adjacent sequence segments. A schematic view of sequence segments scanned by rMAPS2 for the occurrences of consensus motifs in exon skipping events is shown above the respective heatmap. Alternatively regulated exons are depicted as a green box. Only RBPs with a motif enrichment *p*-value (Wilcoxon's rank sum test) <0.001 are shown (related to Fig. 6C). The diamond-shaped labels indicate regions with enriched RBP binding sites (*p*-value < 0.05, Fisher's exact test) from CLIP-seq datasets (Data ref: Zhao et al, 2021). (**D**) Left: Heatmap depicting the temporal enrichment patterns of potential binding sites of class 4 RBPs relative to differentially included exons during the differentiation course (related to Fig. 6E). For each RBP and day-to-day comparison, the minimum enrichment *p*-value across different regions is shown. The diamond-shaped labels highlight the co-occurrence of enriched RBP binding sites (*p*-value < 0.05, Fisher's exact test) and enriched consensus motifs (*p*-value < 0.001, Wilcoxon's rank sum test) in the same region (Data ref: Zhao et al, 2021). Right: Quantification of total gene expression for class 4 RBPs from RNA-seq time-course data. Expression levels were normalized as relative-log expression (RLE) and log10 transformed.

