## [Peer Review File · Molecular Systems Biology]

Uncovering the dynamics and consequences of RNA isoform changes during neuronal differentiation

Andreas Mayer, Jelena Ulicevic, Zhihao Shao, Olga Jasnovidova, Annkatrin Bressin, Martyna Gajos, Alex Ng, Siddharth Annaldasula, David Meierhofer, George Church, and Volker Buskamp

Corresponding author(s): Andreas Mayer (mayer@molgen.mpg.de)

Review Timeline:

Submission Date:	20th Oct 23
Editorial Decision:	29th Nov 23
Revision Received:	27th Feb 24
Editorial Decision:	9th Apr 24
Revision Received:	16th Apr 24
Accepted:	18th Apr 24

Editor: Poonam Bheda

Transaction Report:

29th Nov 2023

Manuscript Number: MSB-2023-12037

Title: Dynamics and consequences of widespread transcript isoform changes during neuronal differentiation

Dear Andreas,

Thank you again for submitting your work to Molecular Systems Biology. We have now heard back from the three reviewers who agreed to evaluate your study. As you will see below, the reviewers appreciate that the proposed approach addresses a timely topic. However, they raise a series of concerns, which we would ask you to address in a major revision.

Without repeating all the issues listed below, some of the more fundamental issues raised by Reviewer 3 are the following:

- the integration of the short-read and long-read sequencing with the proteomics dataset is limited and needs to be improved
- the investigation of RBPs affecting splicing is rather superficial and could be improved by using more confident binding sites

In a cross-commenting session between reviewers and editors, Reviewers 1 and 2 agreed that these points should be improved.

All other issues raised would also need to be satisfactorily addressed. Please note that editorially and in line with Reviewers 1 and 2, we find the novelty sufficient for Molecular Systems Biology. Please let me know in case you would like to discuss in further detail any of the issues raised, I would be happy to schedule a call.

We require:

1) A .docx formatted version of the manuscript text (including legends for main figures, EV figures and tables). Please make sure that the changes are highlighted to be clearly visible. Alternatively you may choose to submit your manuscript as a LaTeX file.

4) A .docx formatted letter INCLUDING the reviewers' reports and your detailed point-by-point responses to their comments. As part of the EMBO Press transparent editorial process, the point-by-point response is part of the Review Process File (RPF), which will be published alongside your paper.

5) A complete author checklist, which you can download from our author guidelines (<https://www.embopress.org/page/journal/17574684/authorguide#submissionofrevisions>). Please insert information in the checklist that is also reflected in the manuscript. The completed author checklist will also be part of the RPF.

6) Please note that all corresponding authors are required to supply an ORCID ID for their name upon submission of a revised manuscript.

7) It is mandatory to include a 'Data Availability' section after the Materials and Methods. Before submitting your revision, primary datasets produced in this study need to be deposited in an appropriate public database, and the accession numbers and database listed under 'Data Availability'. Please remember to provide a reviewer password if the datasets are not yet public (see <https://www.embopress.org/page/journal/17574684/authorguide#dataavailability>).

This study includes no data deposited in external repositories.

8) For data quantification: please specify the name of the statistical test used to generate error bars and P values, the number (n) of independent experiments (specify technical or biological replicates) underlying each data point and the test used to calculate p-values in each figure legend. The figure legends should contain a basic description of n, P and the test applied. Graphs must include a description of the bars and the error bars (s.d., s.e.m.). Please provide exact p values.

9) Our journal encourages inclusion of *data citations in the reference list* to directly cite datasets that were re-used and

obtained from public databases. Data citations in the article text are distinct from normal bibliographical citations and should directly link to the database records from which the data can be accessed. In the main text, data citations are formatted as follows: "Data ref: Smith et al, 2001" or "Data ref: NCBI Sequence Read Archive PRJNA342805, 2017". In the Reference list, data citations must be labeled with "[DATASET]". A data reference must provide the database name, accession number/identifiers and a resolvable link to the landing page from which the data can be accessed at the end of the reference. Further instructions are available at .

<https://www.embopress.org/page/journal/17574684/authorguide#expandedview>

11) For more information: There is space at the end of each article to list relevant web links for further consultation by our readers. Could you identify some relevant ones and provide such information as well? Some examples are patient associations, relevant databases, OMIM/proteins/genes links, author's websites, etc...

12) Author contributions: CRediT has replaced the traditional author contributions section because it offers a systematic machine readable author contributions format that allows for more effective research assessment. Please remove the Authors Contributions from the manuscript and use the free text boxes beneath each contributing author's name in our system to add specific details on the author's contribution. More information is available in our guide to authors.

13) Disclosure statement and competing interests: We updated our journal's competing interests policy in January 2022 and request authors to consider both actual and perceived competing interests. Please review the policy <https://www.embopress.org/competing-interests> and update your competing interests if necessary.

14) Every published paper now includes a 'Synopsis' to further enhance discoverability. Synopses are displayed on the journal webpage and are freely accessible to all readers. They include a short stand first (maximum of 300 characters, including space) as well as 2-5 one-sentences bullet points that summarizes the paper. Please write the bullet points to summarize the key NEW findings. They should be designed to be complementary to the abstract - i.e. not repeat the same text. We encourage inclusion of key acronyms and quantitative information (maximum of 30 words / bullet point). Please use the passive voice. Please attach these in a separate file or send them by email, we will incorporate them accordingly.

Please also suggest a striking image or visual abstract to illustrate your article as a PNG file 550 px wide x 300-600 px high. Share synopsis text and image, as well as eTOC:

Please note that these would be the final versions and changes during proofing are usually not allowed

15) As part of the EMBO Publications transparent editorial process initiative (see our Editorial at <http://embomolmed.embopress.org/content/2/9/329>), Molecular Systems Biology Medicine will publish online a Review Process File (RPF) to accompany accepted manuscripts.

In the event of acceptance, this file will be published in conjunction with your paper and will include the anonymous referee reports, your point-by-point response and all pertinent correspondence relating to the manuscript. Let us know whether you agree with the publication of the RPF and as here, if you want to remove or not any figures from it prior to publication.

Molecular Systems Biology has a "scooping protection" policy, whereby similar findings that are published by others during review or revision are not a criterion for rejection. Should you decide to submit a revised version, I do ask that you get in touch after three months if you have not completed it, to update us on the status.

I look forward to receiving your revised manuscript.

Yours sincerely,

Poonam Bheda, PhD
Scientific Editor
Molecular Systems Biology

Reviewer #1:

In this manuscript Ulicevic et al examine dynamic changes in the splicing isoform profile of iPSCs undergoing neuronal differentiation driven by the induced expression of the neuronal transcription factor Ngn3. RNA-seq analysis revealed distinct gene expression clusters pertinent to specific stages during neuronal differentiation such as stem cells, cell cycle, differentiation, and finally neurons. More than 60% of genes showed isoform switching during differentiation, especially in early and late stages, which occurred through exon skipping, intron retention, alternative start sites or polyadenylation sites. Genes required for neuronal cell development and differentiation were most prone to isoform switching. The authors also used Oxford Nanopore long-read sequencing to capture 10x longer RNA regions, which only partially overlapped with RNA-seq analysis of isoform switching and revealed many non-annotated isoforms. To understand the functional relevance of changes in RNA isoforms, the authors performed mass spectrometry analysis at different timepoints during differentiation, which showed 77 changes in protein isoforms during differentiation and changes in protein expression levels of many RBPs, which may be responsible for alternative splicing. The RNA-binding profile of some of these RBPs matched splice sites relevant for the observed isoform switching events. The experiments were well-executed and the manuscript is very well-written. The data are a valuable resource for the dynamic changes in RNA and protein isoforms during neuronal differentiation.

Major point:

I would like to suggest that the authors highlight the value of integrating different omics data by showcasing two genes that show isoform switching at the RNA and protein level and validating these findings by RT-qPCR and western blotting. One gene could be used as a positive control for which isoform switching/requirement for neuronal differentiation has already been reported.

Another gene could be a new example.

Minor point:

What is the overlap between RNA and protein isoform switching events?

Reviewer #2:

Summary

This study investigates transcript isoform changes in an improved model of neurogenesis from human induced pluripotent stem cells. Using a combination of RNA-seq and full-length (Oxford Nanopore)-seq data, the authors find a strong correlation between shifts in cell morphology and dynamic RNA isoform changes. Some of these changes affect their corresponding protein isoforms. Thousands of stage-specific RNA isoforms are identified, along with potential RNA-binding proteins that drive these changes.

The main findings are the many isoform changes of different types during neuronal differentiation (Figure 3F), of which about 40% have not been previously annotated (Figure 4F), and that 30-45% of them are expressed in a differentiation stage-specific manner (Figure 4G). Many of the isoform changes were predicted to be associated with protein isoform changes, and this was investigated using a complementary Mass spectrometry time series. Due to the difficulty of calling protein isoforms from peptide abundance data, the results are limited to individual cases. Another important finding is that some RNA binding proteins (RBPs), which show stage-specific expression during neuronal differentiation, have binding sites enriched at the positions of alternatively regulated exons.

The presentation of the results is clear and concise, the introduction is leading straight to the point and mentions the relevant recent developments. The data collected by the authors are convincing (sequencing technology, data quality, number of replicates, consistency). The study presented here combines three different omics data sources in a meaningful way to shed light on the transcript (and potentially protein) isoform changes that occur pervasively during neuronal differentiation, suggesting that this is an important regulatory mechanism. The work fits the scope and audience of molecular systems biology. The catalogue of newly discovered isoforms, together with their classification into diverse types of isoform changes, will be a rich resource for the community, and the predicted involvement of RBPs in splicing regulation will lead to testable, mechanistic hypotheses. I therefore consider this work to be a significant advance in the understanding of neuronal differentiation.

My only concerns relate to the quality of the figure captions and the description of the methods, both of which could benefit from further elaboration and comprehensive detail (see comments below)"

Major points

There is an inconsistency between the number of replicates present in the RNA-seq time-series. On page 16, the authors state they conducted 6 biological replicates for RNA-seq. However, Figure 2B and C exhibit only 5 replicates. Kindly explicate this contradiction and provide the rationale behind the potential exclusion replicates.

The experiment's plan could have been better. Although one must acknowledge the experimental effort in producing reproducible results through 5 (or 6) repeats of time series, it would have been more informative to increase the number of sampling time points. My preference would have been three repeated time series with double the number of time points. Quantifying isoforms is a crucial aspect that the manuscript only briefly discusses. Isoform quantification from short read RNA-seq data is notoriously difficult for genes that have many similar isoforms. The authors utilized an algorithm by Li (Bioinformatics

2010). Instead, I anticipated the usage of a more common method such as SALMON (Patro R et al., Nat Methods. 2017). Nevertheless, I acknowledge that there are many options and that overall the DTU analysis has been performed appropriately. There should be a brief discussion of the reliability of isoform quantification. The authors should clarify why they did not carry out a differential transcript expression (DTE) analysis simultaneously. This is important because Fig. 3A illustrates (as the authors mention on p.7) that DTU corresponds with differential gene expression in most instances. Combining DTE and DTU could help identify why the amount of a particular isoform has changed. For example, using both methods could reveal if an alternative promoter was used (visible in both DTE and DTU), or if exon skipping occurred (more likely visible only in DTU). The image captions in general should be more detailed so that the illustrations are more self-contained. I had trouble figuring out some of the graphs (see below).

Minor points

P6: "Interestingly, the first principal component (PC1) that explains 63% of the variation in transcript levels reflects the differentiation time (Fig 2C)". This is not interesting but expected after what has been said before. Please write "Accordingly,..." instead.

P6: Please refer to the Methods when introducing the neural cell differentiation model. Please refer to the Methods to explain how the differentially expressed genes were defined, and how the clustering was performed. The description of the clustering method in the Methods is missing.

Fig. 1 D,E,F: The time series are averaged from 3 replicates, and each time point is plotted as mean plus/minus its standard deviation (which cannot be estimated well from just 3 replicates). It might look less nice but would be more honest to show the three time series separately, for each transcript.

Fig. 2B I suggest showing Spearman correlation instead of Pearson correlation, as Pearson correlation is much more sensitive to outliers and to the normalization method employed. The same applies to Fig. EV3.

Fig. 2E The Gene Ontology analysis is, as in most cases, insignificant. The number of entirely unrelated terms exceeds the number of terms into which one can project a meaning. Why do the authors show only the top 6 (cluster 1) respectively 8 (cluster 4) GO categories (I assume only biological processes were considered) and not all categories which are below the same significance threshold as the one applied in clusters 2 and 3? The barely significant terms in clusters 2 and 3 "by chance" fit well into the picture (developmental maturation, ensheathment of neurons). This has a taste of cherry-picking. Leave this subfigure out.

Fig. EV1. Subfigures A,B: The vertical color bar at the left of the heatmap is not annotated. The colors seem to indicate clusters of different dynamic behavior. Please clarify and explain. Additionally, it must be described how "differentially expressed" was assessed/tested in a 5-point time series. In Fig. EV1 A, please ensure that the gene set you picked ("KEGG-CELL-CYCLE") are either genes that are periodically expressed during the cell cycle, or that they are either known to enhance respectively inhibit cell cycle progression. It would be informative to know the cell cycle phase in which the differentially expressed genes are typically active.

Subfigure C: It has not been stated from which source the bona fide marker genes for different neuronal cell types were obtained.

Subfigure D: It does not make sense to z-transform Spearman correlation values of the whole brain-regions times time points map, as it merely hides the information about the absolute value of the correlation, without adding anything to the plot. I suggest keeping the absolute values and simple re-scaling the color bar. (cool colors for values below the overall average, warm colors for higher correlation values).

Figure EV3: A: Please indicate whether the correlations were calculated on the gene level or on the isoform level. Use the more robust Spearman correlation, as done in Main Figure 4D,E. Then, please compare the ONT-based Spearman correlations in Fig. 2B to EV3A. Ideally, EV3A will show a higher correlation between replicates and a lower correlation between days than Fig 2B.

Subfigure D: I do not understand the caption: "Schematic view of isoform types". Which isoform types? Subfigure C: How do the annotated isoforms inferred by RNA-seq respectively ONT-seq look like, how can one explain the difference? Why do you trust RNA-seq based predictions at all if they are so unreliable? Subfigure E: The coverage plot below is enigmatic to me, and its description is insufficient. Which of the sequenced reads were assigned to the annotated isoform, which of them were assigned to the postulated novel isoform? Why don't we see a full coverage of the retained intron in the reads pertaining to the novel isoform? In the main text, it is said that the novel isoform is lacking a region at the C-terminus. This region is not visible to me, nor is it properly highlighted.

I found it hard to comprehend Fig 4D, E. The text states that the RNA-seq and ONT-seq samples were compared. Assuming there were 5 duplicates per day, comparing all-day versus day-to-day data should result in a 5x3 table. However, as opposed to what was stated in the methods, there appear to be only 4 RNA-seq duplicates on day 3.

Fig. 5A: The representation of the numbers of genes with colored areas does not make sense, as the genes with a domain loss, say between day 0 and day 3 do not necessarily appear again in the comparison between day 3 and day 5, specifically most likely not in the domain loss category. Therefore, a simple stacked barplot for day 0 vs. 3 and another stacked barplot for day 3 vs. 5 is appropriate.

I was missing the mapping statistics in the Methods, both for RNA-seq and ONT-seq. How many reads could be mapped to transcripts per sample (average, min, max)?

The manuscript titled "Dynamics and consequences of widespread transcript isoform changes during neuronal differentiation" presents an analysis of two transcriptomic and a proteomic dataset along a time-course of in vitro neuronal differentiation of human iPS cells.

While the differentiation model exhibits some interesting features, and the generated datasets could serve as a valuable resource for researchers, the analyses presented in this manuscript represent a very preliminary stage of investigation. Moreover, the manuscript lacks a clear and compelling biological question that drives the research. Additionally, the central finding of the article - that distinct cell states during differentiation involve alterations in RNA alternative splicing patterns - is well-established and has been extensively characterized in previous studies.

*Specific Concerns

1. Unclear Aim: The manuscript's title and abstract emphasize the "dynamics" of splicing patterns, yet the study falls short of providing a comprehensive assessment of alternative splicing (or cleavage/poly-adenylation) regulation dynamics. Instead, it focuses on characterizing steady-state levels of transcriptome isoforms at different stages of a differentiation protocol.

2. Lack of Novelty: The manuscript fails to present novel findings. The time-points evaluated correspond to distinct cell states (stem cells, neuronal precursors, immature neurons), and it is expected that these states exhibit differences in their transcript repertoire at both gene and isoform levels. This is well-established knowledge, and the underlying mechanisms have been thoroughly characterized. Transcriptome data on neuronal differentiation of pluripotent cell cultures have been available for years (e.g., 10.1073/pnas.0914114107). Similarly, the key roles of alternative splicing regulation in this process are well-documented, as acknowledged by the authors themselves (p. 3, last paragraph).

3. Ineffective Integration of Datasets: Despite the potential benefits of combining the short-read and long-read RNA-seq data with mass spectrometry data, the authors have not successfully integrated these datasets to gain deeper insights.

- Short-read vs. Long-read RNA-seq: The advantages of using long-read sequencing to characterize complete transcript isoforms are well-recognized by molecular biologists. However, the authors devote significant time and an entire main figure (Fig. 3) to analyzing the results of a "quick and dirty" method for characterizing isoform changes from short-read RNA-seq. The value of this analysis is unclear, given the inherent limitations of identifying whole isoforms from short-read data. The authors themselves acknowledge these limitations in Fig. 4 (particularly 4E, showing intra-condition correlations below 0.5) and EV3 (specifically EV3C, demonstrating that isoforms predicted from short-read data are vastly different from those obtained with long-read data). No further attempts are made to combine the two datasets for analyzing isoform regulation.

- Splicing Regulator Expression: The authors' analysis of splicing regulator expression, intended to provide mechanistic insights, is superficial and fails to yield any novel findings. It is unclear why gene expression quantification was not performed using the short-read data, which typically offer greater depth. Additionally, while the authors claim to investigate RBP binding sites near regulated exons, their analysis is isoform-centered rather than event-centered, raising questions about the origin of this set of regulated exons. They also rely on predicted binding sites, leading to a non-specific prediction of numerous possible RBPs. A co-analysis of gene expression of these RBPs and splicing regulation is not performed, nor is there any attempt to utilize published RBP binding data (such as iCLIP) for these proteins.

- Proteomic Data: The proteomic data is analyzed entirely independently of the transcriptomic data. The authors merely state that they observe differences in isoforms at the RNA and protein levels, a conclusion that is again not novel in any aspect. However, they make no attempt to link the two analyses, for instance, by verifying the isoform changes detected with the ONT data on the proteomic dataset (at least for the small subset of proteins where informative differential peptides can be detected). In conclusion, the inclusion of this dataset does not significantly enhance the gene expression characterization presented in the preceding figures.

Point-by-point response to reviewers

We thank the reviewers for their thorough evaluation of our manuscript and their highly constructive comments. A detailed point-by-point response to all comments and suggestions of the reviewers is provided in this document. Based on the reviewer's critiques we have performed a set of new computational analyses, and re-wrote parts of the manuscript. Notably, we have conducted new integrative analysis of the RNA-seq, ONT-seq and proteomics time course data. We have also improved our RBP analysis by integration of available CLIP-seq data that helped us to identify RBP binding sites in proximity to alternative exons with greater confidence. The obtained new results led to new data panels that we included in the revised manuscript: Fig 2E; Fig 3G; Fig 5F-J; Fig 6A-E; Fig EV1D; Fig EV2B; Fig EV4A-F; Fig EV5A-D, Appendix Figure S1-S3. We believe that the new data and clarifications that we added to the main text have significantly strengthened this work. Again, we thank the reviewers for their thoughtful assessment of our work. The full point-by-point response is provided below (blue letters).

Reviewer #1:

In this manuscript Ulicevic et al examine dynamic changes in the splicing isoform profile of iPSCs undergoing neuronal differentiation driven by the induced expression of the neuronal transcription factor Ngn3. RNA-seq analysis revealed distinct gene expression clusters pertinent to specific stages during neuronal differentiation such as stem cells, cell cycle, differentiation, and finally neurons. More than 60% of genes showed isoform switching during differentiation, especially in early and late stages, which occurred through exon skipping, intron retention, alternative start sites or polyadenylation sites. Genes required for neuronal cell development and differentiation were most prone to isoform switching. The authors also used Oxford Nanopore long-read sequencing to capture 10x longer RNA regions, which only partially overlapped with RNA-seq analysis of isoform switching and revealed many non-annotated isoforms. To understand the functional relevance of changes in RNA isoforms, the authors performed mass spectrometry analysis at different timepoints during differentiation, which showed 77 changes in protein isoforms during differentiation and changes in protein expression levels of many RBPs, which may be responsible for alternative splicing. The RNA-binding profile of some of these RBPs matched splice sites relevant for the observed isoform switching events. The experiments were well-executed and the manuscript is very well-written. The data are a valuable resource for the dynamic changes in RNA and protein isoforms during neuronal differentiation.

We thank the reviewer for the thorough evaluation of our work and the thoughtful comments.

Major point:

I would like to suggest that the authors highlight the value of integrating different omics data by showcasing two genes that show isoform switching at the RNA and protein level and validating these findings by RT-qPCR and western blotting. One gene could be used as a positive control for which isoform switching/requirement for neuronal differentiation has already been reported. Another gene could be a new example.

We thank the reviewer for this suggestion. Based on this comment and critiques of Reviewer 3, we performed a set of new integrative computational analyses. By integrating short-read (Illumina) RNA-seq, long-read (ONT) RNA-seq and the proteomics time course data sets, we uncovered RNA isoform changes that underlie corresponding protein isoform trend changes with high confidence. For example, we could detect the switch in the expression of the two known RNA isoforms of *PFN2*, *PFN2a* and *PFN2b*, that have been reported in previous studies (for example: PMID: 34458253). *PFN2* can serve as a positive control and we now use it as a prime example in our manuscript (Fig 3B, Fig 5B, G, I, Fig EV4D and Appendix Fig S2). However, our new analysis also revealed new gene examples that undergo RNA- and protein-isoform changes during the differentiation course. We now show *MACF1* and *SMARCB1*, two genes that have been linked to neuronal diseases. For both genes we could detect a clear switch in the expression of two RNA isoforms by Illumina RNA-seq and ONT-seq, as well as a corresponding trend change of the two corresponding protein isoforms by the proteomics data. Notably, this analysis also uncovered new expressed non-annotated protein isoforms as for example in case of *MACF1*. The results of the new analyses are shown in the following data panels of the revised manuscript: Fig 5I, J, EV4D, E, F and Appendix Fig S2 (and previous Figures 3B and 5B).

The high reproducibility of the RNA-seq Illumina time course data (4 to 5 replicate time course experiments) as well as of the independent ONT-seq experiments (3 replicate time course experiments), together with the corresponding highly reproducible proteomics time course data sets (3 replicate time course experiments) allowed us to reveal examples for dynamic isoform changes during neuronal differentiation with greater confidence. Given all the new integrative analyses, we think that RT-qPCR experiments that rely on a small number of primer pairs and other inherent limitations such as the requirement of a reference gene, whose expression remains constant during the differentiation course, would not significantly increase the confidence level of our findings. Furthermore, antibodies specifically targeting the different *PFN2* protein isoforms as well as for the newly detected *MACF1* protein isoforms that can be used in Western blot experiments are not commercially available. We believe that the new results highlight the importance and the power of data integration, and further strengthen the main conclusions of this work.

Minor point:

What is the overlap between RNA and protein isoform switching events?

To address the reviewer's question, we first checked how RNA changes generally translate into protein changes, and second, how RNA isoform changes translate into protein isoforms specifically. We observed high Spearman correlations (0.5-0.6) between RNA and protein changes for the same differentiation stages (new Fig EV4A). This correlation increased considerably (to 0.8-0.9) when considering only genes with significant changes during the differentiation course (new Fig 5F and EV4B). This analysis also revealed that RNA changes from the previous differentiation stage correlated equally well with protein level changes (new Fig EV4A and new Fig. 5F). This new integrative correlation analysis suggests a persistent manifestation of RNA changes into protein level changes, of up to two days. The same trend was detected when we compared long-read ONT RNA expression and protein levels. We observed high Spearman correlations (0.8-0.9) between genes with a significant change in RNA levels during the differentiation course (new Fig EV4B). We next speculated whether the observed persistent effect also holds true for the changes in protein isoforms. Indeed, the protein isoform-specific correlation analysis revealed similar

effects measured by short- (new Fig EV4C) and long-read RNA-seq (new Fig. 5H). Our new results suggest that RNA changes from the previous two days can underlie protein isoform changes and the observed trend differences.

Reviewer #2:

Summary

This study investigates transcript isoform changes in an improved model of neurogenesis from human induced pluripotent stem cells. Using a combination of RNA-seq and full-length (Oxford Nanopore)-seq data, the authors find a strong correlation between shifts in cell morphology and dynamic RNA isoform changes. Some of these changes affect their corresponding protein isoforms. Thousands of stage-specific RNA isoforms are identified, along with potential RNA-binding proteins that drive these changes.

The main findings are the many isoform changes of different types during neuronal differentiation (Figure 3F), of which about 40% have not been previously annotated (Figure 4F), and that 30-45% of them are expressed in a differentiation stage-specific manner (Figure 4G). Many of the isoform changes were predicted to be associated with protein isoform changes, and this was investigated using a complementary Mass spectrometry time series. Due to the difficulty of calling protein isoforms from peptide abundance data, the results are limited to individual cases. Another important finding is that some RNA binding proteins (RBPs), which show stage-specific expression during neuronal differentiation, have binding sites enriched at the positions of alternatively regulated exons.

The presentation of the results is clear and concise, the introduction is leading straight to the point and mentions the relevant recent developments. The data collected by the authors are convincing (sequencing technology, data quality, number of replicates, consistency). The study presented here combines three different omics data sources in a meaningful way to shed light on the transcript (and potentially protein) isoform changes that occur pervasively during neuronal differentiation, suggesting that this is an important regulatory mechanism. The work fits the scope and audience of molecular systems biology. The catalogue of newly discovered isoforms, together with their classification into diverse types of isoform changes, will be a rich resource for the community, and the predicted involvement of RBPs in splicing regulation will lead to testable, mechanistic hypotheses. I therefore consider this work to be a significant advance in the understanding of neuronal differentiation.

My only concerns relate to the quality of the figure captions and the description of the methods, both of which could benefit from further elaboration and comprehensive detail (see comments below)"

We thank the reviewer for the thorough evaluation of our work and the thoughtful comments.

Major points

There is an inconsistency between the number of replicates present in the RNA-seq time-series. On page 16, the authors state they conducted 6 biological replicates for RNA-seq. However, Figure 2B and C exhibit only 5 replicates. Kindly explicate this contradiction and provide the rationale behind the potential exclusion replicates.

We thank the reviewer for pointing this out and apologize for this inconsistency. Data was generated in 5 replicates for Day 0 to Day 5. One of the replicates of Day 3 appeared as an

outlier during data processing (low mapping of reads as compared to other samples) and was excluded from further analyses. We added clarification and corrected the manuscript accordingly.

The experiment's plan could have been better. Although one must acknowledge the experimental effort in producing reproducible results through 5 (or 6) repeats of time series, it would have been more informative to increase the number of sampling time points. My preference would have been three repeated time series with double the number of time points.

We agree with the reviewer's advice. This is a great suggestion and we will consider this in upcoming time series experiments.

Quantifying isoforms is a crucial aspect that the manuscript only briefly discusses. Isoform quantification from short read RNA-seq data is notoriously difficult for genes that have many similar isoforms. The authors utilized an algorithm by Li (Bioinformatics 2010). Instead, I anticipated the usage of a more common method such as SALMON (Patro R et al., Nat Methods. 2017). Nevertheless, I acknowledge that there are many options and that overall the DTU analysis has been performed appropriately. There should be a brief discussion of the reliability of isoform quantification.

Based on this comment, we conducted RNA isoform quantification using Salmon v.1.9.0 (PMID: 28263959) in alignment-based mode and compared the results with those obtained from RSEM v.1.2.25 (PMID: 21816040). We computed the Spearman's correlation coefficients to assess the agreement between TPM values estimated from RSEM and Salmon across identical samples for filtered (median TPM >1) genes (n=5,197) and isoforms (n=31,104), respectively. The median of obtained correlation values is 0.96 for genes and 0.90 for isoforms (Rebuttal Figure 1, below this response). Our comparative analyses indicate the comparable performance between RSEM and Salmon in terms of both gene-level and isoform-level quantification. These findings highlight the reproducibility of RNA isoform quantification.

RSEM and Salmon share similar methodologies. For instance, both RSEM and Salmon in alignment-mode require reads to be aligned directly to the transcriptome rather than to the genome. Additionally, both tools utilize iterations of the EM algorithm for quantification (PMID: 28263959).

Furthermore, our observations are consistent with previous benchmarking studies using simulated and real data, which underscored the strong concordance between isoform quantification results obtained from RSEM and Salmon in both alignment-based and mapping-based modes. These studies suggest that both Salmon and RSEM are viable options for quantifying RNA isoforms with comparable effectiveness (PMID: 34034652, PMID: 24885830, PMID: 28784092, PMID: 26201343, PMID: 27107712).

Rebuttal Figure 1: The box plot representation depicts the distribution of Spearman's correlation coefficients, which measure the agreement between TPM values estimated from RSEM and Salmon across identical samples, for filtered genes and isoforms, respectively.

The authors should clarify why they did not carry out a differential transcript expression (DTE) analysis simultaneously. This is important because Fig. 3A illustrates (as the authors mention on p.7) that DTU corresponds with differential gene expression in most instances. Combining DTE and DTU could help identify why the amount of a particular isoform has changed. For example, using both methods could reveal if an alternative promoter was used (visible in both DTE and DTU), or if exon skipping occurred (more likely visible only in DTU).

We thank the reviewer for this comment. The DTU analysis quantifies the relative contribution of individual transcripts to total gene expression, highlighting shifts in transcript usage patterns. This distinguishes DTU from DTE, which assesses individual transcript levels, independently of other co-expressed transcripts of the respective gene. To compare the results of DTE and DTU within our datasets, we performed a DTE analysis using DESeq2 with transcript-level expression in RNA-seq (fold change >1.5 and adjusted p-value <0.05 as significant DTE). Our findings revealed a significantly higher number of isoforms exhibiting DTE but no DTU detected (33,327 out of 48,915) compared to isoforms with detected DTU but no DTE (8,432 out of 23,930) (Fisher's exact test p-value <2.2e-16) (Rebuttal Figure 2, below this response). For example, transcript ENST00000417582.6 of *GATAD2A* became the predominant transcript after day 2 without showing significant changes in individual expression levels (Rebuttal Figure 3, below this response). In contrast, although transcript ENST00000501597.3 of *RPL41* exhibited significant DTE throughout neurogenesis, this would not lead to any notable alterations in isoform or protein functionality since it was the only expressed transcript (Rebuttal Figure 4, below this response). Thus, by prioritizing DTU over DTE, we can effectively filter out genes that predominantly express a single transcript and focus on alterations in the composition of isoform expression. Notably, our study investigated alternative splicing and functional consequences at the protein-level of pairs of isoforms with large opposing changes in isoform usage during neurogenesis, a task unattainable through the analysis of individual isoforms alone.

Nevertheless, we acknowledged the importance of DTE for the analyses on a per-transcript basis, especially for the unannotated novel transcript isoforms detected in our ONT-seq data. We conducted a DTE analysis to identify transcript isoforms with differential expression

between days in ONT-seq. This approach offered valuable insights into the expression patterns of newly identified isoforms at different stages of neuronal differentiation. We now mention the results of this analysis in the results section “Nanopore sequencing reveals new RNA isoforms at all stages of neurogenesis”. We added a description of this analysis to the methods section of the revised manuscript.

Rebuttal Figure 2: Venn diagram of transcripts with DTE and DTU between any two days of the differentiation course.

Rebuttal Figure 3: Expression of *GATAD2A* and its top four expressed transcript isoforms. Example of isoforms with detected DTU but no DTE.

Rebuttal Figure 4: Expression of *RPL41* and its top four expressed transcript isoforms. Example of isoforms exhibiting DTE but no DTU detected.

The image captions in general should be more detailed so that the illustrations are more self-contained. I had trouble figuring out some of the graphs (see below).

Minor points

P6: "Interestingly, the first principal component (PC1) that explains 63% of the variation in transcript levels reflects the differentiation time (Fig 2C)". This is not interesting but expected after what has been said before. Please write "Accordingly,..." instead.

Done as suggested by the reviewer.

P6: Please refer to the Methods when introducing the neural cell differentiation model. Please refer to the Methods to explain how the differentially expressed genes were defined, and how the clustering was performed. The description of the clustering method in the Methods is missing.

We thank the reviewer for pointing this out. We now refer to the methods section when we introduce the neuronal cell differentiation model. We also added a description of the clustering method and we now point the reader to the methods section ("RNA-seq data analysis" under the subsection entitled "Differential gene expression analysis"), when we mention the differential gene expression and the cluster analysis the first time.

Fig. 1 D,E,F: The time series are averaged from 3 replicates, and each time point is plotted as mean plus/minus its standard deviation (which cannot be estimated well from just 3 replicates). It might look less nice but would be more honest to show the three time series separately, for each transcript.

We now provide additional plots for each marker gene separately below this response, showing all individual replicate measurements (Rebuttal Figure 5). As the measurements are highly similar, we have kept the original median plots in Figure 1 for simplicity. We have updated the panel for neuronal markers, due to a minor visual correction of the x axis, and also added additional clarifications to the corresponding methods section entitled "RT-qPCR and RT-qPCR data analysis".

Rebuttal Figure 5: Expression of pluripotency (*OCT4*, *NANOG*, *SOX2*), neural progenitor (*NES*, *NOTCH1*), and neuronal markers (*DCX*, *MAP2*, *TUBB3*) measured by RT-qPCR. Each plot represents one marker gene expression with each biological replicate measurement

plotted individually and indicated as a single dot (n=3) as suggested by the reviewer. Additionally, the mean of the log2 transformed fold change (n=3) is plotted as a bar, and scale bars represent the respective standard deviation.

Fig. 2B I suggest showing Spearman correlation instead of Pearson correlation, as Pearson correlation is much more sensitive to outliers and to the normalization method employed. The same applies to Fig. EV3.

We have performed new correlation analyses for our RNA-seq and ONT-seq time course data sets, now using Spearman correlation. Both correlation analyses led to highly similar results, with slightly higher correlations between biological replicates in case of Pearson correlation, as shown in the heatmap representations below this response (Rebuttal Figures 6 and 7). This is in line with previous observations for medium-sized datasets (30-100), providing evidence that Pearson correlation analysis performs slightly better for RNA-seq data (PMID: 35089979).

Rebuttal Figure 6: Left: Pearson correlation analysis as shown in Fig 2B. Right: New Spearman correlation analysis as suggested by the reviewer.

Rebuttal Figure 7: Left: Pearson correlation analysis as shown in Fig EV3A. Right: New Spearman correlation analysis as suggested by the reviewer.

Fig. 2E The Gene Ontology analysis is, as in most cases, insignificant. The number of entirely unrelated terms exceeds the number of terms into which one can project a meaning. Why do the authors show only the top 6 (cluster 1) respectively 8 (cluster 4) GO categories (I assume only biological processes were considered) and not all categories which are below the same significance threshold as the one applied in clusters 2 and 3? The barely significant terms in clusters 2 and 3 "by chance" fit well into the picture (developmental maturation, ensheathment of neurons). This has a taste of cherry-picking. Leave this subfigure out.

We thank the reviewer for pointing this out. Based on the reviewer comment, we have revised the analysis and performed a GO term enrichment analysis for the differentially expressed genes ($p_{adj} < 0.05$ and $|\log_2FC| > 1$) of subsequent days. We updated Fig 2E accordingly and now show all GO-slim terms with an FDR ≤ 0.01 in the new Appendix Figure S1. If GO terms are hierarchically clustered, we depict the most specific GO term exclusively. In main Figure 2, however, we present only the top 10 candidates due to space restrictions. At the beginning of the differentiation course, Day0-Day1, we mainly obtained developmental-related GO terms. In later stages of the neuronal cell differentiation time course, for example Day3-Day5, we observed an enrichment of more neuron-related terms such as axon guidance and synapse organization (new Fig 2E and new Appendix Figure S1).

Fig. EV1. Subfigures A,B: The vertical color bar at the left of the heatmap is not annotated. The colors seem to indicate clusters of different dynamic behavior. Please clarify and explain. Additionally, it must be described how "differentially expressed" was assessed/tested in a 5-point time series. In Fig. EV1 A, please ensure that the gene set you picked ("KEGG-CELL-CYCLE") are either genes that are periodically expressed during the cell cycle, or that they are either known to enhance respectively inhibit cell cycle progression. It would be informative to know the cell cycle phase in which the differentially expressed genes are typically active.

In Figure EV1A, Cluster 1 and 2 include active genes in the mitotic phase (from day 0 to day 2) and the post-mitotic phase (from day 3 to day 5), respectively. In Figure EV1B, Cluster 1, 2, and 3 include transcription factors that were primarily expressed on day 0, from day 1 to day 3, and on day 5, respectively. We added clarifications to the legend of Figure EV1A and B.

Genes that showed differential expression in at least one pairwise day-to-day comparison were considered for the analysis related to Figure EV1A. We added the explanation to the corresponding figure legend.

The human gene set "KEGG_CELL_CYCLE" from the Molecular Signatures Database contains 125 key genes known to be involved in mitotic cell cycle regulation including genes that are required for cell cycle progression such as cyclin-dependent kinases (CDKs), transcription factors, cyclin-CDK inhibitors (CKIs) and cell cycle checkpoint kinases.

Subfigure C: It has not been stated from which source the bona fide marker genes for different neuronal cell types were obtained.

We apologize for this lack of information. The marker genes were based on the "Neural markers guide" published by the biotechnology company Abcam (<https://docs.abcam.com/pdf/neuroscience/neural-markers-guide-web.pdf>). The neural

markers were all based on experimental evidence (please see references in the guide). We now added this information to the legend of EV1C.

Subfigure D: It does not make sense to z-transform Spearman correlation values of the whole brain-regions times time points map, as it merely hides the information about the absolute value of the correlation, without adding anything to the plot. I suggest keeping the absolute values and simple re-scaling the color bar. (cool colors for values below the overall average, warm colors for higher correlation values).

We thank the reviewer for pointing this out. We now plot the absolute values of the Spearman's correlation coefficients (maximum: 0.71, minimum: 0.62, and median: 0.68) and set the median of correlation values as the midpoint in the color scale of the new Figure EV1D.

Figure EV3: A: Please indicate whether the correlations were calculated on the gene level or on the isoform level. Use the more robust Spearman correlation, as done in Main Figure 4D,E. Then, please compare the ONT-based Spearman correlations in Fig. 2B to EV3A. Ideally, EV3A will show a higher correlation between replicates and a lower correlation between days than Fig 2B. Subfigure D: I do not understand the caption: "Schematic view of isoform types". Which isoform types? Subfigure C: How do the annotated isoforms inferred by RNA-seq respectively ONT-seq look like, how can one explain the difference? Why do you trust RNA-seq based predictions at all if they are so unreliable? Subfigure E: The coverage plot below is enigmatic to me, and its description is insufficient. Which of the sequenced reads were assigned to the annotated isoform, which of them were assigned to the postulated novel isoform? Why don't we see a full coverage of the retained intron in the reads pertaining to the novel isoform? In the main text, it is said that the novel isoform is lacking a region at the C-terminus. This region is not visible to me, nor is it properly highlighted.

Figure EV3A shows Pearson's correlation on the gene level. We added clarification to the new figure legend of EV3A. We show the Spearman's correlation of gene expression for ONT-seq data in Rebuttal Figure 7. By comparing Rebuttal Figures 6 and 7, we indeed observe lower between-day correlations for the ONT-seq as compared to short RNA-seq data.

Figure EV3D depicts the main types of transcript isoforms that were identified from ONT-seq data and classified based on their agreement with the intron structure of the reference transcript by SQANTI3. We included a detailed explanation of each category in the revised figure legend of EV3D.

Rebuttal Figure 8 shows the transcript isoforms of *MMP23B* inferred by RNA-seq and detected by ONT-seq, respectively. TCONS_00000255 corresponds to ENST00000356026.9 (ISM), and TCONS_00000266 corresponds to ENST00000356026.9 (FSM). A potential reason why the isoform expression profiles from RNA-seq and ONT-seq were so different is that ONT-seq can capture transcript isoforms in full length and covers multiple consecutive splice junctions in a single read (Figure 4B), whereas short reads from RNA-seq typically cover only one splice junction (if at all) and have higher mapping uncertainty due to their relatively short length. In addition, *MMP23B* is located within a tandemly duplicated genomic region of chromosome 1p36.3 (PMID: 9750192). As a result, short reads from RNA-seq may not be uniquely mapped, while long reads from ONT-seq can encompass the entire transcripts and therefore may map uniquely. *MMP23B* represents rather a special case due to its duplicated nature, so we should not discard isoform inference from short-read RNA-seq since we observed a relatively high

correlation of isoform-level expressions between RNA-seq and ONT-seq (Figure 4E). Instead, we should acknowledge the advantage of long-read ONT-seq.

The number of reads that mapped to the novel isoform and annotated isoform of the third replicate of day 5 (the sample shown in EV3E) is 77 and 94, respectively. Of note, the quantification is based on the alignment to the transcriptome while EV3E depicts the alignment to the reference genome.

One possible explanation for the incomplete coverage of the retained intron in certain reads is that long reads may not be mapped contiguously. As a result, minimap2 may split the alignment into multiple segments, leading to supplementary alignments, a phenomenon commonly observed in Nanopore sequencing (PMID: 31366910).

We added Appendix Figure S3 to show the loss of the oligonucleotide/oligosaccharide binding (OB)-fold domain at the C-terminus in the novel *DHX15* isoform “TCONS00087410” and updated the main text accordingly.

Rebuttal Figure 8: Top: Transcript isoforms of *MMP23B* in GENCODE v28 annotation for RNA-seq data. Below: Transcript isoforms of *MMP23B* as identified in ONT-seq data.

I found it hard to comprehend Fig 4D, E. The text states that the RNA-seq and ONT-seq samples were compared. Assuming there were 5 duplicates per day, comparing all-day versus day-to-day data should result in a 5x3 table. However, as opposed to what was stated in the methods, there appear to be only 4 RNA-seq duplicates on day 3.

We thank the reviewer for pointing this out. As mentioned in our response to the first major comment of this reviewer, we generated data for 5 biological replicates for Day 0 to Day 5. However, one of the replicates of Day 3 appeared as an outlier during data processing (low

mapping of reads as compared to other samples) and was excluded from all further analyses. We added clarification to the main text of the manuscript and apologize for this inconsistency.

Fig. 5A: The representation of the numbers of genes with colored areas does not make sense, as the genes with a domain loss, say between day 0 and day 3 do not necessarily appear again in the comparison between day 3 and day 5, specifically most likely not in the domain loss category. Therefore, a simple stacked barplot for day 0 vs. 3 and another stacked barplot for day 3 vs. 5 is appropriate.

We have changed Figure 5A by adding a stacked barplot as suggested by the reviewer.

I was missing the mapping statistics in the Methods, both for RNA-seq and ONT-seq. How many reads could be mapped to transcripts per sample (average, min, max)?

The rates of primary mapped reads for RNA-seq have a mean value of 0.878, a minimum of 0.768 and a maximum of 0.906. The rates of primary mapped reads for ONT-seq have a mean value of 0.729, a minimum of 0.597 and a maximum of 0.865. We added this information to the corresponding methods section of RNA-seq and ONT-seq, respectively.

Reviewer #3:

The manuscript titled "Dynamics and consequences of widespread transcript isoform changes during neuronal differentiation" presents an analysis of two transcriptomic and a proteomic dataset along a time-course of in vitro neuronal differentiation of human iPS cells.

While the differentiation model exhibits some interesting features, and the generated datasets could serve as a valuable resource for researchers, the analyses presented in this manuscript represent a very preliminary stage of investigation. Moreover, the manuscript lacks a clear and compelling biological question that drives the research. Additionally, the central finding of the article - that distinct cell states during differentiation involve alterations in RNA alternative splicing patterns - is well-established and has been extensively characterized in previous studies.

We thank the reviewer for the thorough evaluation of our work and the thoughtful comments.

*Specific Concerns

1. Unclear Aim: The manuscript's title and abstract emphasize the "dynamics" of splicing patterns, yet the study falls short of providing a comprehensive assessment of alternative splicing (or cleavage/poly-adenylation) regulation dynamics. Instead, it focuses on characterizing steady-state levels of transcriptome isoforms at different stages of a differentiation protocol.

The main goal of this systems-wide study was to uncover the dynamic changes of transcript isoforms and the functional consequences, and to provide insights into the determinants of isoform changes during human neuronal cell differentiation. We now adapted the title of the

manuscript to better reflect the active goal of this work. Furthermore, the main objectives should also become clear from the abstract and the last paragraph of the introduction section.

Our title and abstract (and other sections of our manuscript) refer to the 'dynamics' of transcript isoform changes which means changes in the abundance of transcript isoforms over time during the differentiation time course. In this regard, we use the term in accordance with previous studies on RNA isoform changes, for instance: PMID: 20194744; PMID: 20436464.

Our thorough comparison of transcript isoform abundance and identities between time points that includes several biological replicate measurements revealed strong and reproducible dynamic changes during the differentiation course, clearly going beyond steady-state level profiling at individual differentiation time points as stated by the reviewer. Results and visualizations of the comparative analyses are in nearly all main figures of the manuscript.

2. Lack of Novelty: The manuscript fails to present novel findings. The time-points evaluated correspond to distinct cell states (stem cells, neuronal precursors, immature neurons), and it is expected that these states exhibit differences in their transcript repertoire at both gene and isoform levels. This is well-established knowledge, and the underlying mechanisms have been thoroughly characterized. Transcriptome data on neuronal differentiation of pluripotent cell cultures have been available for years (e.g., 10.1073/pnas.0914114107). Similarly, the key roles of alternative splicing regulation in this process are well-documented, as acknowledged by the authors themselves (p. 3, last paragraph).

We disagree with this opinion. We provide several lines of novel findings in our study: First, we present a new human cell differentiation model that we have established and that allows the *in vitro* analysis of human neurogenesis under defined conditions, and in standard stem cell media. We also show that this differentiation model is amenable for genome-wide analyses because cells transition synchronously and efficiently through the different cell stages during the differentiation time course. Second, we identified thousands of new transcript isoforms and transcript isoform changes providing a more complete view on RNA isoforms during neuronal differentiation. Third, we characterized the functional consequences of RNA abundance- and RNA isoform-changes. During the revision, we performed a set of new integrative analyses for the transcriptome- and proteome-wide time course data. The results of the new analyses indicate that RNA isoform changes can underlie differentiation stage-specific protein-isoform changes. The analysis also identified new expressed protein isoforms. Fourth, our study reveals underlying mechanisms of the widespread RNA isoform changes and their relative contributions to the diversification of the transcriptome during human neuronal differentiation, also including the alternative usage of transcription start and polyadenylation sites. Fifth, our systems-wide study sheds light on the determinants of isoform changes. Specifically, our results point to the importance of RNA binding proteins that are expressed in a differentiation stage-specific manner.

The study from Wu et al. 2010 to which the reviewer is referring to represents an important initial work on the dynamics of transcript isoform changes during human neuronal cell differentiation. Although this early work reveals extensive transcript isoform changes during neurogenesis which is consistent with our findings, the functional consequences of the RNA isoform changes and the determinants remained unknown. Furthermore, this study exclusively focused on RNA isoforms that emerged from alternative splicing not appreciating other mechanisms that contribute to the complexity of isoforms such as the usage of alternative

transcription start and poly-adenylation sites. Therefore, the mechanisms of isoform changes during human neurogenesis remained to a large extent unclear. Finally, this study uses an *in vitro* differentiation model that relies on the addition and withdrawal of growth factors from the growth media to enable differentiation which can impact the efficiency and also how synchronously cells transition through different developmental phases, and often results in a heterogeneous population of different cell types. Along these lines, it was not clear to what extent immature neuron- or glia-like cells were obtained. We now included this pioneer study on transcript isoforms during neuronal differentiation in our list of references.

3. Ineffective Integration of Datasets: Despite the potential benefits of combining the short-read and long-read RNA-seq data with mass spectrometry data, the authors have not successfully integrated these datasets to gain deeper insights.

- Short-read vs. Long-read RNA-seq: The advantages of using long-read sequencing to characterize complete transcript isoforms are well-recognized by molecular biologists. However, the authors devote significant time and an entire main figure (Fig. 3) to analyzing the results of a "quick and dirty" method for characterizing isoform changes from short-read RNA-seq. The value of this analysis is unclear, given the inherent limitations of identifying whole isoforms from short-read data. The authors themselves acknowledge these limitations in Fig. 4 (particularly 4E, showing intra-condition correlations below 0.5) and EV3 (specifically EV3C, demonstrating that isoforms predicted from short-read data are vastly different from those obtained with long-read data). No further attempts are made to combine the two datasets for analyzing isoform regulation.

The motivation of our in-depth comparison of short- and long-read RNA-seq data was to identify and quantify similarities and potential differences in the detection of RNA isoforms and isoform changes using the same neuronal cell differentiation model. To our knowledge a thorough comparison of state-of-the-art Illumina short-read and Oxford nanopore (ONT) long-read RNA-seq that also includes sufficient replicate measurements to enable a proper statistical analysis in a cell differentiation context is still lacking. Furthermore, several current studies on RNA processing still use short-read RNA-seq data to call RNA isoforms ignoring the potential advantages of long-read RNA-seq methods. Therefore, we believe that our results of the comparative analysis that we show in Figure 4 and the corresponding Figure EV3 will be helpful for the RNA and differentiation communities.

Based on the reviewer comment, we have performed a new set of integrative computational analysis for the ONT-seq, RNA-seq and MS time course data. We show the main results of the new analyses in Fig 5 F, G, H, I, J, Fig 6A, and EV4A, B, C, D, E, F. The new integrative analysis of RNA-seq and ONT-seq data for the expression of RBPs shows that the dynamic changes of RBP expression during differentiation appear method-independent (new Fig 6A). Consistent with our initial analysis, we observed increased expression of RBPs at day five that are associated with "nervous system development" and "cytoskeleton organization" (new Fig 6B). The new integrative approach further revealed an enrichment of RBPs linked to alternative splicing regulation which is also in line with our finding that the majority of isoform switches occurred at day 5 (Fig 3C), further strengthening our conclusions.

-Splicing Regulator Expression: The authors' analysis of splicing regulator expression, intended to provide mechanistic insights, is superficial and fails to yield any novel findings. It is unclear why gene expression quantification was not performed using the short-read data,

which typically offer greater depth. Additionally, while the authors claim to investigate RBP binding sites near regulated exons, their analysis is isoform-centered rather than event-centered, raising questions about the origin of this set of regulated exons. They also rely on predicted binding sites, leading to a non-specific prediction of numerous possible RBPs. A co-analysis of gene expression of these RBPs and splicing regulation is not performed, nor is there any attempt to utilize published RBP binding data (such as iCLIP) for these proteins.

We have now also performed the expression analysis of RBPs with the short-read Illumina RNA-seq data sets and observed similar expression changes as in case of ONT-seq data sets (new Fig 6A).

The alternative transcription and splicing analysis was run in two phases, integrating both isoform-centric and event-centric analyses. First, IsoformSwitchAnalyzeR performed the differential transcript usage analysis (DTU) and splicing classification based on full-length transcript isoforms. This step globally revealed the number of alternative transcription and splicing events between days of the differentiation course. Second, rMATS performed an event-level differential splicing analysis, specifically targeting genes exhibiting DTU identified by IsoformSwitchAnalyzeR. The set of regulated exons identified by rMATS were filtered to capture all significant differential splicing events (FDR <0.05) within genes exhibiting a significant DTU (FDR <0.05 and dIF >0.1) to increase robustness. We have included a detailed description in the methods section of the revised manuscript.

Furthermore, we have integrated publicly available CLIP-seq data to identify RBP binding sites in proximity to alternative exons with greater confidence. This new analysis revealed a set of RBPs, including HuR, for which the corresponding consensus RNA binding motif was enriched in the vicinity of the alternative exon and that can be bound by the corresponding RBP (CLIP-seq binding signal). Our RNA-seq time course data further showed that the corresponding RBPs are indeed expressed revealing sets of RBPs that may underlie the observed isoform changes at different phases of the differentiation course. The new results are shown in Fig 6C, D, E and EV5 A, B, C, D.

- Proteomic Data: The proteomic data is analyzed entirely independently of the transcriptomic data. The authors merely state that they observe differences in isoforms at the RNA and protein levels, a conclusion that is again not novel in any aspect. However, they make no attempt to link the two analyses, for instance, by verifying the isoform changes detected with the ONT data on the proteomic dataset (at least for the small subset of proteins where informative differential peptides can be detected). In conclusion, the inclusion of this dataset does not significantly enhance the gene expression characterization presented in the preceding figures.

Based on this comment and on the suggestion of Reviewer 1, we performed new computational analyses to integrate the RNA-seq (short and long-read) and the proteomics time course data sets.

From the RNA isoform transcriptome (long-read RNA-seq data), we first built our new reference proteome. The advantage of this strategy is that it also allows the identification of non-annotated protein isoforms that are expressed and new protein isoform changes. This new analysis allowed us to identify 76 proteins that undergo clear protein isoform trend changes during the differentiation course (absolute log₂FC difference ≥1) (new Fig 5G). Next, we checked how RNA changes generally translate into protein changes, and second, how

RNA isoform changes translate into protein isoforms specifically. We observed high Spearman correlations (0.5-0.6) between RNA and protein changes considering identical differentiation stages (new Fig EV4A). This correlation increases considerably (to 0.8-0.9) when considering only genes with significant changes across the differentiation time (new Fig 5F and new Fig EV4B). This analysis also revealed that RNA changes from the previous differentiation stage correlate equally well with protein level changes (new Fig EV4A, B and new Fig 5F). This observation suggests a persistent or delayed effect of RNA changes on the protein level of up to two days.

We next speculated whether the observed delay also holds true for the changes in protein isoforms. Indeed, the protein isoform-specific correlation analysis revealed similar delays measured by short- (new Fig EV4C) and long-read RNA-seq (new Fig 5H). Our new results suggest that RNA isoform changes from the previous two days can underlie protein isoform changes and the observed trend differences.

For example, we detected the switch in the expression of the two known RNA isoforms of PFN2, PFN2a and PFN2b, that have been reported in previous studies (for instance: PMID: 34458253). However, our newly performed analysis also revealed new examples that undergo RNA- and protein-isoform changes during the differentiation course. We now show MACF1 and SMARCB1, two genes that have been linked to neuronal diseases. For both genes we could detect a clear switch in the expression of two RNA isoforms by RNA-seq and ONT-seq, as well as a corresponding trend change of the two corresponding protein isoforms by MS. Notably, this analysis revealed the expression of new non-annotated protein isoforms as for example in case of MACF1 (Fig 5G, I, J and EV4D, E, F). We think that the new analyses further strengthen our conclusion that RNA isoform changes can underlie protein isoform changes during human neuronal differentiation.

9th Apr 2024

Manuscript Number: MSB-2023-12037R

Title: Uncovering the dynamics and consequences of RNA isoform changes during neuronal differentiation

Dear Dr Mayer,

Thank you for the submission of your revised manuscript to Molecular Systems Biology. We have now received the enclosed reports from the referees that were asked to re-assess it. As you will see the reviewers are now globally supportive and I am pleased to inform you that we will be able to accept your manuscript pending the following final amendments and appropriate response to reviewers:

- 1) In the main manuscript file, please reduce keywords to max. 5.
- 2) Please reformat the Data Availability statement according to the example below:
"The datasets and computer code produced in this study are available in the following databases:
- Chip-Seq data: Gene Expression Omnibus GSE46748 (<https://www.ncbi.nlm.nih.gov/geo/query/acc.cgi?acc=GSE46748>)
- Modeling computer scripts: GitHub (<https://github.com/SysBioChalmers/GECKO/releases/tag/v1.0>)
- [data type]: [full name of the resource] [accession number/identifier] [(doi or URL or identifiers.org/DATABASE:ACCESSION)]"
- 3) Data availability: Please ensure that the sequencing and proteomics data are now publicly available.
- 4) Code: Please include a README file on Github with practical use instructions for potential future users of your code.
- 5) Please rename "Conflict of Interest" to "Disclosure and competing interests statement". We updated our journal's competing interests policy in January 2022 and request authors to consider both actual and perceived competing interests. Please review the policy <https://www.embopress.org/competing-interests> and update your competing interests if necessary.
- 6) Data not shown: We do not allow statements/conclusions with "data not shown". As per our guidelines, on "Unpublished Data" the journal does not permit citation of "Data not shown". All data referred to in the paper should be displayed in the main or Expanded View figures. Please remove from page 36.
- 7) In the Materials and Methods, please take care of the following:
 - Cell lines: Please include all information requested in the author checklist for cell lines used in the manuscript (accession number in repository or supplier name, catalog number, clone number, and/or RRID). Currently there is no information on how this cell line was obtained. Please also be sure to include a sentence in the Materials and Methods as to whether or not the cell lines were recently authenticated.
 - Please ensure that a statement on whether or not blinding was done is included in the Materials and Methods even if no blinding was done.
- 8) Primers: Instead of having the table of primers in Table EV1 please move the table to the Appendix and rename as Appendix Table S1. Please also ensure that the legend is moved to the Appendix and that the callout to this table is included in the main manuscript text. Please also update the Author Checklist to indicate that primer information can be found in the Appendix.
- 9) Please ensure that the individual sections of the manuscript are in the following order: Title page - Abstract & Keywords - Introduction - Results - Discussion - Materials & Methods - Data Availability - Acknowledgements - Disclosure and Competing Interests Statement - References - Figure Legends - Expanded View Figure Legends.
- 10) For the figures and figure legends, please take care of the following:
 - Figure Legends (main + EV): Please note that a separate 'Data Information' section is required in the legends of figures 1d-f.
 - Please note that the box plots need to be defined in terms of minima, maxima, centre, bounds of box and whiskers, and percentile in the legends of figures 5i-j; EV 4d-f.
 - Please note that information related to n is missing in the legend of figure 4b.
 - Please note that the error bar is not defined in the legend of figure 4b.
 - Please note that the scale bar needs to be defined for figure 1c.
- 11) Appendix file: Please add the manuscript title on the table of contents page.
- 12) Synopsis:
 - Synopsis image: Please upload the synopsis image as a high-resolution jpeg file 550 pixels wide x (250-400) pixels high.
 - Synopsis text: Please shorten the standfirst to maximum 300 characters, including spaces.
 - Please check your synopsis text and image before submission with your revised manuscript. Please be aware that in the proof stage minor corrections only are allowed (e.g., typos).
- 13) Source Data: Source data files need to be saved in a scheme with one figure per folder and then uploaded as .zip files - e.g. all the Source data files for figure 1 need to be saved in a single folder (with each file clearly labeled for the corresponding figure panel) and this needs to be zipped and then uploaded as "SD figure 1.zip" file.
- 14) As part of the EMBO Publications transparent editorial process initiative (see our policy here: https://www.embopress.org/transparent-process#Review_Process), Molecular Systems Biology will publish online a Peer Review File (PRF) to accompany accepted manuscripts. This file will be published in conjunction with your paper and will include the anonymous referee reports, your point-by-point response and all pertinent correspondence relating to the manuscript. Let us know whether you agree with the publication of the PRF and as here, if you want to remove or not any figures from it prior to publication. Please note that the Authors checklist will be published at the end of the PRF.
- 15) Please provide a point-by-point letter INCLUDING my comments as well as the reviewer's reports and your detailed

responses (as Word file).

I look forward to reading a new revised version of your manuscript as soon as possible.

Sincerely,

Poonam

Poonam Bheda, PhD
Scientific Editor
Molecular Systems Biology

Please click on the link below to submit your revised manuscript:

Reviewer #1:

The authors have adequately addressed my points and included additional analyses that reveal the novel aspects and findings of their study.

Reviewer #2:

The authors have improved the manuscript considerably. They have added a substantial amount of information in the Methods section and in the expanded view figures. The authors also performed a second isoform quantification using Salmon and demonstrated a high degree of agreement with their RSEM method (Rebuttal Figure 1). An additional differential transcript expression analysis was performed and mentioned in the manuscript. All minor points I raised have been addressed to my full satisfaction.

I have merely one remark the authors can easily address. In the rebuttal letter, the authors say that one replicate at day 3 has been removed from the analysis due to quality issues. However, I could not find a clarifying statement in the revised manuscript (neither in the Results, Methods, nor Legend of Fig. 2B). Please add it.

Finally, I found one type on p18: "[...] (Langmead et al, 2009)he resulting alignments were employed [...]" - after the closing bracket it should read " the" and not "he".

Reviewer #3:

The authors have made a significant effort to improve the manuscript in response to the reviewers' comments. Most of my specific criticisms have been addressed.

Notably, the integrative analysis of the three datasets has been significantly expanded, resulting in a much more valuable resource for the community. In addition, the RBP motif analysis has been complemented by available CLIP data, which undoubtedly aids the interpretation of the data presented here.

As a result of the authors' efforts, the manuscript is now of much higher quality and suitable for publication.

While I still consider that the paper lacks novel biological insights, I see that the characterization of the cell culture neuronal differentiation model and the combined datasets generated here can be very valuable resources for many researchers.

Minor criticisms:

The following are some minor points that the authors may want to consider:

1. The introduction could be condensed to improve readability.
2. There may be an inaccuracy in the diagram representing alternative transcription start sites (ATSS) in Fig. 3E (first drawing). The concern is that alternative first exons cannot be entirely used as second exons when an upstream TSS is used through splicing. This is because the 5' end, defined as the transcription start site, is unlikely to contain the consensus sequences for a 3'

splice site.

For the vast majority of ATSS cases, either the first exons of each TSS ("red" or "black" in the figure) are completely absent from the other isoform (similar to the shown ATTS, but in the 5' end), or the upstream TSS starts slightly before the downstream TSS, resulting in an extended first exon containing both "red" and "black" exons (not connected through splicing).

Point-by-point response to the editor and reviewers

We thank the editor and the reviewers for their constructive feedback on our manuscript. All comments and suggestions helped us to further improve the manuscript. A detailed point-by-point response to all comments and suggestions is provided below (blue letters).

1) In the main manuscript file, please reduce keywords to max. 5.

We reduced the number of key words to 5 as requested.

2) Please reformat the Data Availability statement according to the example below:

"The datasets and computer code produced in this study are available in the following databases:

- Chip-Seq data: Gene Expression Omnibus GSE46748
(<https://www.ncbi.nlm.nih.gov/geo/query/acc.cgi?acc=GSE46748>)

- Modeling computer scripts: GitHub
(<https://github.com/SysBioChalmers/GECKO/releases/tag/v1.0>)

- [data type]: [full name of the resource] [accession number/identifier] ([doi or URL or identifiers.org/DATABASE:ACCESSION)]"

We have reformatted the Data Availability section accordingly.

3) Data availability: Please ensure that the sequencing and proteomics data are now publicly available.

Done.

4) Code: Please include a README file on Github with practical use instructions for potential future users of your code.

We moved the repository to our institute server and added a README file describing the methods and tools used and the file structure of the project.

5) Please rename "Conflict of Interest" to "Disclosure and competing interests statement". We updated our journal's competing interests policy in January 2022 and request authors to consider both actual and perceived competing interests. Please review the policy <https://www.embopress.org/competing-interests> and update your competing interests if necessary.

Done.

6) Data not shown: We do not allow statements/conclusions with "data not shown". As per our guidelines, on "Unpublished Data" the journal does not permit citation of "Data not shown". All data referred to in the paper should be displayed in the main or Expanded View figures. Please remove from page 36.

We have removed "not shown" from the legend of Figure 6A.

7) In the Materials and Methods, please take care of the following:

- Cell lines: Please include all information requested in the author checklist for cell lines used in the manuscript (accession number in repository or supplier name, catalog number, clone number, and/or RRID). Currently there is no information on how this cell line was obtained. Please also be sure to include a sentence in the Materials and Methods as to whether or not the cell lines were recently authenticated.

We have provided additional information on cell line generation, repository IDs, and the name of the supplier of the cell lines used, where available. We have also added additional information about cell line authentication.

- Please ensure that a statement on whether or not blinding was done is included in the Materials and Methods even if no blinding was done.

We have also included a statement about blinding as requested.

8) Primers: Instead of having the table of primers in Table EV1 please move the table to the Appendix and rename as Appendix Table S1. Please also ensure that the legend is moved to the Appendix and that the callout to this table is included in the main manuscript text. Please also update the Author Checklist to indicate that primer information can be found in the Appendix.

We have added the table of primers as Appendix Table S1 and changed the main text and author checklist accordingly.

9) Please ensure that the individual sections of the manuscript are in the following order: Title page - Abstract & Keywords - Introduction - Results - Discussion - Materials & Methods - Data Availability - Acknowledgements - Disclosure and Competing Interests Statement - References - Figure Legends - Expanded View Figure Legends.

We have checked that the sections are in the correct order.

10) For the figures and figure legends, please take care of the following:

- Figure Legends (main + EV): Please note that a separate 'Data Information' section is required in the legends of figures 1d-f.

We have updated the legends of Figure 1D-F, followed by a separate "Data Information" section.

- Please note that the box plots need to be defined in terms of minima, maxima, centre, bounds of box and whiskers, and percentile in the legends of figures 5i-j; EV 4d-f.

We have included the definition of box plots in the legends of Figures 5I-J; EV 4D-F.

- Please note that information related to n is missing in the legend of figure 4b.

We have now included information in the legend of Figure 4B on the number of short and long reads spanning junctions used to calculate the percentage.

- Please note that the error bar is not defined in the legend of figure 4b.

We have added the definition of the error bar, which represents the standard error of the mean to the legend of Figure 4B.

- Please note that the scale bar needs to be defined for figure 1c.

We have added the definition of the scale bar representing the standard deviation to the legend of Figure 1C.

11) Appendix file: Please add the manuscript title on the table of contents page.

Done.

12) Synopsis:

- Synopsis image: Please upload the synopsis image as a high-resolution jpeg file 550 pixels wide x (250-400) pixels high.

Done.

- Synopsis text: Please shorten the standfirst to maximum 300 characters, including spaces.

We have shortened the standfirst of the synopsis as requested. It is now 287 characters long.

Done.

13) Source Data: Source data files need to be saved in a scheme with one figure per folder and then uploaded as .zip files - e.g. all the Source data files for figure 1 need to be saved in a single folder (with each file clearly labeled for the corresponding figure panel) and this needs to be zipped and then uploaded as "SD figure 1.zip" file.

We have added the Source Data folder as requested.

14) As part of the EMBO Publications transparent editorial process initiative (see our policy here: https://www.embopress.org/transparent-process#Review_Process), Molecular Systems Biology will publish online a Peer Review File (PRF) to accompany accepted manuscripts. This file will be published in conjunction with your paper and will include the anonymous referee reports, your point-by-point response and all pertinent correspondence relating to the manuscript. Let us know whether you agree with the publication of the PRF and as here, if you want to remove or not any figures from it prior to publication. Please note that the Authors checklist will be published at the end of the PRF.

We agree.

15) Please provide a point-by-point letter INCLUDING my comments as well as the reviewer's reports and your detailed responses (as Word file).

Done.

Reviewer #1:

The authors have adequately addressed my points and included additional analyses that reveal the novel aspects and findings of their study.

We thank the reviewer for the constructive and supportive comments throughout the review process.

Reviewer #2:

The authors have improved the manuscript considerably. They have added a substantial amount of information in the Methods section and in the expanded view figures. The authors also performed a second isoform quantification using Salmon and demonstrated a high degree of agreement with their RSEM method (Rebuttal Figure 1). An additional differential transcript expression analysis was performed and mentioned in the manuscript. All minor points I raised have been addressed to my full satisfaction.

We thank the reviewer for the constructive and supportive comments throughout the review process.

I have merely one remark the authors can easily address. In the rebuttal letter, the authors say that one replicate at day 3 has been removed from the analysis due to quality issues. However, I could not find a clarifying statement in the revised manuscript (neither in the Results, Methods, nor Legend of Fig. 2B). Please add it.

We apologize and have now added this statement to the Materials and Methods section.

Finally, I found one typo on p18: "[...] (Langmead et al, 2009)he resulting alignments were employed [...]" - after the closing bracket it should read " the" and not "he".

We apologize and have corrected the typo.

Reviewer #3:

The authors have made a significant effort to improve the manuscript in response to the reviewers' comments. Most of my specific criticisms have been addressed.

Notably, the integrative analysis of the three datasets has been significantly expanded, resulting in a much more valuable resource for the community. In addition, the RBP motif analysis has been complemented by available CLIP data, which undoubtedly aids the interpretation of the data presented here.

As a result of the authors' efforts, the manuscript is now of much higher quality and suitable for publication.

While I still consider that the paper lacks novel biological insights, I see that the characterization of the cell culture neuronal differentiation model and the combined datasets generated here can be very valuable resources for many researchers.

We thank the reviewer for the constructive and helpful comments throughout the review process.

Minor criticisms:

The following are some minor points that the authors may want to consider:

1. The introduction could be condensed to improve readability.

Given the interdisciplinary nature of our study, we believe it is important to provide sufficient background on the various key topics, including cell differentiation, neuronal differentiation models, RNA isoform dynamics and determinants, and methods. However, based on the reviewer's comment, we have reduced the text wherever possible and systematically further improved the readability of the introductory section.

2. There may be an inaccuracy in the diagram representing alternative transcription start sites (ATSS) in Fig. 3E (first drawing). The concern is that alternative first exons cannot be entirely used as second exons when an upstream TSS is used through splicing. This is because the 5' end, defined as the transcription start site, is unlikely to contain the consensus sequences for a 3' splice site.

For the vast majority of ATSS cases, either the first exons of each TSS ("red" or "black" in the figure) are completely absent from the other isoform (similar to the shown ATTS, but in the 5' end), or the upstream TSS starts slightly before the downstream TSS, resulting in an extended first exon containing both "red" and "black" exons (not connected through splicing).

We agree with the reviewer and have changed Figure 3E accordingly.

18th Apr 2024

Manuscript number: MSB-2023-12037RR

Title: Uncovering the dynamics and consequences of RNA isoform changes during neuronal differentiation

Dear Dr Mayer,

Thank you again for sending us your revised manuscript. We are now satisfied with the modifications made and I am pleased to inform you that your paper has been accepted for publication.

Yours sincerely,

Sincerely,

Poonam Bheda, PhD
Scientific Editor
Molecular Systems Biology
